# `LIVEditor-14B`:
# Lightning Unified Video Editing via In-Context Sparse Attention

**Shitong Shao** [*1]   **Zikai Zhou** [*1]   **Haopeng Li** [1]   **Yingwei Song** [2]   **Wenliang Zhong** [1]   **Lichen Bai** [1]   **Zeke Xie** [†1]

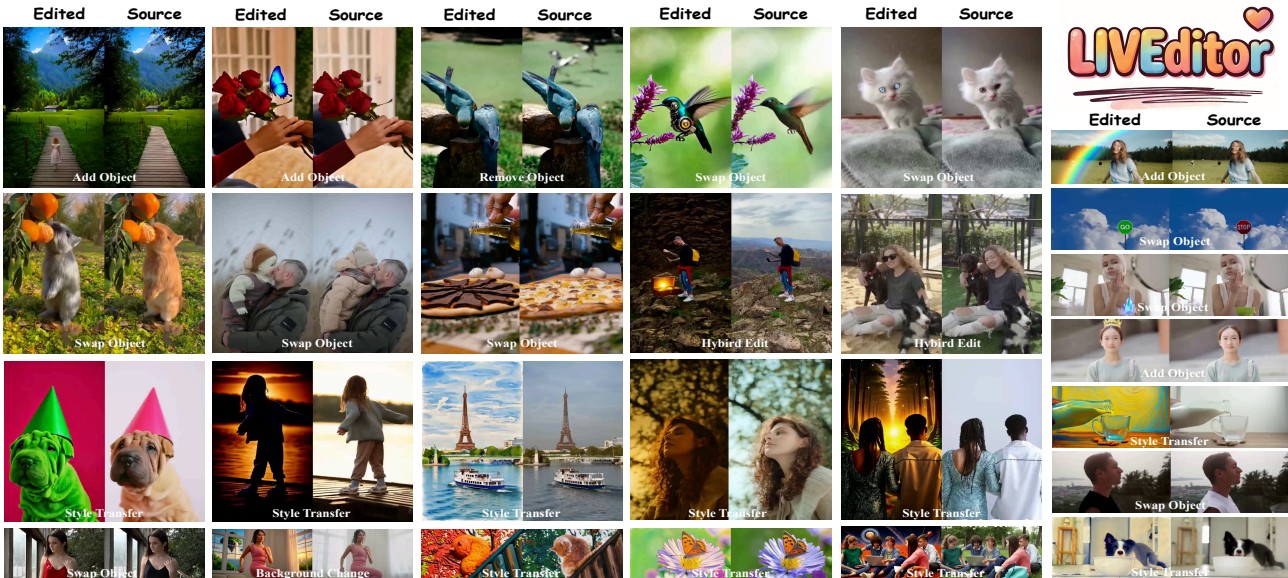

*Figure 1.* **Visualization of `LIVEditor-14B` (ISA).** The superior video editing performance of `LIVEditor-14B` (ISA) stems from a unified framework that leverages in-context sparse attention.

## Abstract

Video editing has evolved toward In-Context Learning (ICL) paradigms, yet the resulting quadratic attention costs create a critical computational bottleneck. In this work, we propose **I**n-context **S**parse **A**ttention (**ISA**), the first near-lossless empirical sparse framework tailored for ICL video editing. Our design is grounded in two key insights: ***first***, context tokens exhibit significantly lower saliency than source tokens; ***second***, we theoretically prove and empirically validate that Query sharpness correlates with approximation error. Motivated by these findings, ISA implements an efficient *pre-selection* strat-
egy to prune redundant context, followed by a dynamic *query grouping* mechanism that routes high-error queries to full attention and low-error ones to a computationally efficient 0-th order Taylor sparse attention. Furthermore, we build `LIVEditor-14B`, a novel lightning video editing model via ISA and a proposed video-editing data pipeline that curated a 1.7M high-quality dataset. Extensive experiments demonstrate that `LIVEditor-14B` achieves a $\sim$60% reduction in attention-module latency while surpassing state-of-the-art methods across EditVerseBench, IVE-Bench, and VIE-Bench, delivering near-lossless acceleration without compromising visual fidelity. Our code is released in LIVEditor-14B.

*Equal contribution [1]Hong Kong University of Science and Technology (GZ), Hong Kong [2]University of Arizona, USA. Correspondence to: Zeke Xie <zekexie@hkust-gz.edu.cn>.

*Proceedings of the $43^{rd}$ International Conference on Machine Learning*, Seoul, South Korea. PMLR 306, 2026. Copyright 2026 by the author(s).

## 1. Introduction

Recent vision foundation models (Chen et al., 2025a; Google, 2025a;b; Kong et al., 2024; Shou, 2024; WanTeam

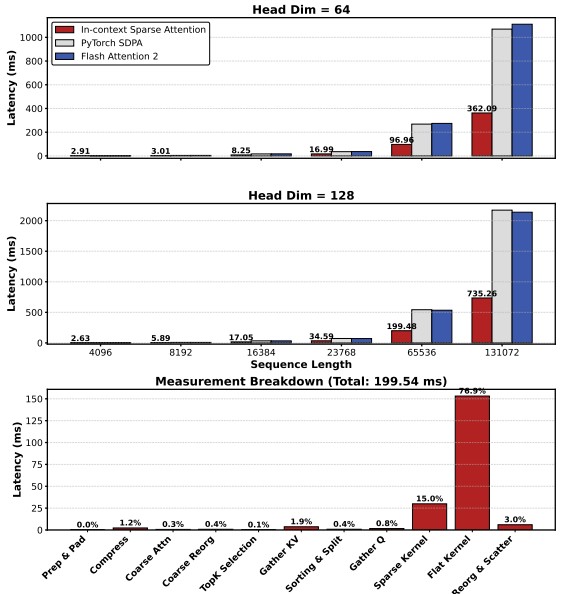

*Figure 2.* The speedup of ISA relative to SDPA and FA2 becomes increasingly pronounced as sequence length grows. Within the ISA framework, the computational cost is dominated by the sparse kernel (0-order Taylor attention) and the flat kernel (full-attn), while the overhead of remaining operations is negligible.

et al., 2025; Wu et al., 2025) are increasingly unified and applied to downstream tasks. While image editing (Cai et al., 2025; Google, 2025a) is mature, video editing is rapidly advancing from domain-specific methods (Ju et al., 2023; Zhang et al., 2023) to unified frameworks (Liang et al., 2025; Yu et al., 2025) (Bai et al., 2025; Ju et al., 2025; Mou et al., 2025). Simultaneously, architectures are shifting from complex cross-attention mechanisms (Jiang et al., 2025; Lee et al., 2025; Zhang et al., 2024; Zi et al., 2025c) toward scalable In-Context Learning (ICL) paradigms (Ju et al., 2025; Mou et al., 2025; Wei et al., 2025; Ye et al., 2025), which maximize information assimilation by directly concatenating context and source tokens via full attention.

**Challenge.** However, in video generation, attention mechanisms represent the primary computational bottleneck due to the inherent long-sequence characteristics of video data. As the sequence length scales from 5K to 50K, the computational cost increases quadratically with sequence length. This limitation is further exacerbated by attention in ICL. Specifically, in video editing tasks, the number of context tokens is typically commensurate with the number of source tokens, quadrupling the computational cost and consequently leading to substantial increases in GPU memory usage and latency. Most existing sparse attention mechanisms (Li et al., 2025b; Zhang et al.; 2025a;b) are designed based on general video generation and fail to account for the distinction between context tokens and source tokens, thus underutilizing the specific characteristics of the ICL scenario to design efficient and high-performance sparse attention mechanisms.

**Contribution.** In this work, we address the critical absence of efficient sparse attention mechanisms for ICL through a systematic investigation that bridges theoretical insights with practical application.

- **Key Finding.** We first revisit the attention mechanism in ICL contexts through a rigorous distribution analysis. Our investigation reveals a pivotal observation: context tokens typically contribute a negligible proportion to the total attention score, indicating limited saliency. This suggests that a vast majority of context tokens can be effectively pruned without compromising representational fidelity, provided the most critical tokens are retained.
- **Theoretical Analysis.** We theoretically demonstrate that Query sharpness is proportional to the approximation error of the 0-th order Taylor expansion. This establishes Query sharpness as an efficient indicator for dynamic grouping: high-sharpness queries demand precise computation to preserve fidelity, whereas low-sharpness queries can be safely approximated to save cost.
- **ISA.** We propose **I**n-context **S**parse **A**ttention (**ISA**), an experimentally lossless attention for video editing. ISA first retains critical tokens via a pre-selection strategy, and then innovatively employs a novel grouping mechanism: high-error queries utilize full attention for fine-grained features, while low-error queries use our block-wise 0-th order Taylor sparse attention. This method approximates interactions via tiled Key-Value means, reducing complexity from $\mathcal{O}(N^2 D)$ to $\mathcal{O}(N^2 D/b)$.
- **LIVEditor-14B.** We build **LIVEditor-14B**, an experimentally lossless lightning video editing model, via ISA and a proposed video-editing data pipeline. The proposed data pipeline constructed a massive dataset comprising over 1.7M high-quality video editing pairs, systematically categorized into tasks such as style transfer, object swapping, and human editing, generated via a comprehensive automated pipeline involving VLMs and diffusion models. LIVEditor-14B demonstrates the effectiveness of ISA and our data pipeline at scale.
- **Empirical Success.** Our experiments demonstrate that LIVEditor-14B achieves near-lossless acceleration and superior performance. **First**, ISA reduces attention-module latency by approximately 60% compared to standard SDPA and FlashAttention v2/3 (Fig. 2). **Second**, LIVEditor-14B consistently outperforms state-of-the-art models (e.g., Ditto (Bai et al., 2025), InsV2V (Cheng et al., 2023), Lucy Edit (Team, 2025)) across benchmarks like EditVerseBench (Ju et al., 2025) and VIE-Bench (Mou et al., 2025). **Finally**, ISA generalizes effectively to training-free settings, accelerating pre-trained models without degradation and offering significantly better visual quality than alternative sparse mechanisms.

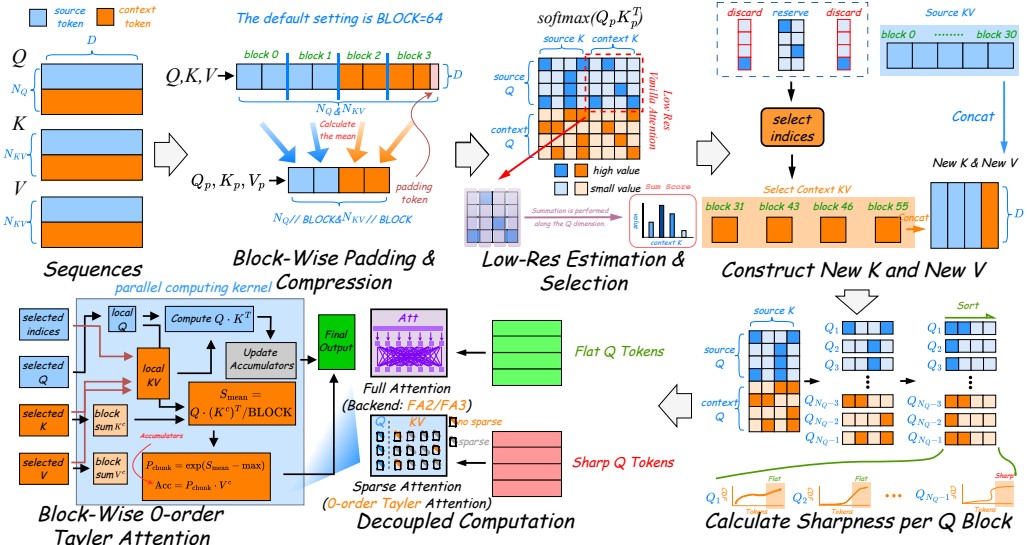

*Figure 3.* **The workflow of In-Context Sparse Attention (ISA)**. ISA optimizes efficiency by enforcing sparsity across both Query and Key/Value dimensions. The process begins by identifying and retaining only the most salient Key and Value pairs from the context tokens. Subsequently, Queries are partitioned based on a computed sharpness metric. Queries exhibiting high sharpness undergo full attention computation, while flat Queries, characterized by minimal Taylor expansion errors, are processed using an accelerated block-wise 0-order Taylor attention mechanism.

## 2. Preliminary

**Standard Attention.** A general attention mechanism processes the following inputs: a Query tensor $Q \in \mathbb{R}^{B \times H \times N \times D}$, a Key tensor $K \in \mathbb{R}^{B \times H \times S \times D}$, and a Value tensor $V \in \mathbb{R}^{B \times H \times S \times D}$, where $B$, $H$, $N$, $S$, and $D$ denote the batch size, number of attention heads, sequence length of the Queries, sequence length of the Keys, and head dimension, respectively. The mechanism first computes the score matrix as $S_{QK} = QK^\top / \sqrt{D}$, subsequently derives the attention weights via $P_{QK} = \text{softmax}(S_{QK})$, and finally produces the output $O = P_{QK}V$.

**Block Sparse Attention.** Sparsification of standard attention is typically achieved by reducing the effective size of $K$ and $V$, i.e., by selecting a subset of indices from $S$ for computation. Early approaches employed element-wise sparse attention by defining a binary mask $M \in \{0, 1\}^{N \times S}$ and pruning computations via $S_{QK} = M \odot S_{QK}$ (where $\odot$ denotes the Hadamard product). However, this unstructured sparsity pattern is generally ill-suited for hardware acceleration. A more hardware-efficient alternative is to operate at the block level. For video models, the model first forms contiguous spatiotemporal tiles, then flattens these tiles in tile order as a sequence. We then partition this sequence into non-overlapping blocks, denoted as $\{Q_i \in \mathbb{R}^{B \times H \times L_Q \times D}\}_{i=1}^{N_Q}$, $\{K_j \in \mathbb{R}^{B \times H \times L_K \times D}\}_{j=1}^{N_K}$, and $\{V_j \in \mathbb{R}^{B \times H \times L_K \times D}\}_{j=1}^{N_K}$. Here, $L_Q$ and $L_K$ represent the block sizes, while $N_Q = \lceil N/L_Q \rceil$ and $N_K = \lceil S/L_K \rceil$ denote the number of Query and Key/Value blocks, respectively. Under this framework, the block mask assumes a shape of $N_Q \times N_K$, where $M_{ij} = 0$ indicates that the com-

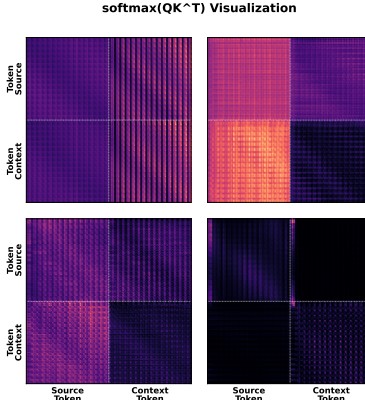

*Figure 4.* In ICL, the attention matrix exhibits distinct distributional characteristics, manifesting as four discernible regions. This structural pattern suggests that attention mechanisms should be specifically tailored for ICL scenarios.

putation of the attention scores $Q_i K_j^\top$ and the subsequent aggregation $(P_{QK})_{ij} V_j$ are bypassed.

**Pooling Attention.** Efficiently determining the binary values of $M_{ij}$ necessitates a lightweight selection mechanism. Pooling attention (Zhang et al.; 2025a) substantiates to be a highly suitable solution for this purpose. As illustrated in the "*Block-Wise Padding & Compression*" part of Fig. 3, pooling attention first applies pooling along the sequence dimension to yield the compressed representations $Q^c \in \mathbb{R}^{B \times H \times N_K \times D}$, $K^c, V^c \in \mathbb{R}^{B \times H \times N_K \times D}$. Because this sequence derives from the spatial-temporal tile ordering used in video encoders, the coarse representations preserve local structure while still being hardware-friendly. Standard attention is then performed on these tensors, effectively

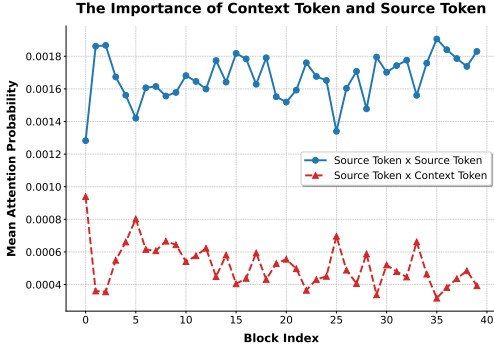

*Figure 5.* In ICL, attention scores between source Queries and source Keys are typically significantly higher than those between source Queries and context Keys. This disparity becomes increasingly pronounced in deeper layers.

reducing the computational complexity from $\mathcal{O}(NSD)$ to $\mathcal{O}(N_Q N_K D)$. Finally, a Top-$k$ selection strategy is applied to the attention map $S_{\text{coarse}} = \text{softmax}(Q^c(K^c)^\top)$ to derive the block mask $M_{\text{coarse}} \in \mathbb{R}^{B \times H \times N_Q \times N_K}$.

## 3. In-Context Sparse Attention

In this section, we first detail *pre-selection* employed by ISA to identify critical context tokens. Subsequently, we introduce *block-wise 0-th order Taylor sparse attention*, an efficient algorithm for attention approximation. We then analyze the correlation between Query sharpness and the Taylor approximation error. Finally, we present *grouped computation*, which executes attention operations of varying complexity across distinct Query groups.

**Motivation of *Pre-Selection*.** The principal distinctions between full attention in ICL and that in general video generation tasks are twofold: **(1)** the token count is effectively doubled, and **(2)** the tokens exhibit a distinct structural division into source and context tokens, which are stored contiguously in hardware memory. A fundamental question arises: do context tokens and source tokens contribute equally to the attention in video editing? To investigate this, we visualize the attention score matrix in Fig. 4. The distributions of the four interaction patterns—source Query $Q^{\text{src}} \times$ source Key $(K^{\text{src}})^\top$, context Query $Q^{\text{ctx}} \times$ context Key $(K^{\text{ctx}})^\top$, cross-term $Q^{\text{src}}(K^{\text{ctx}})^\top$, and cross-term $Q^{\text{ctx}}(K^{\text{src}})^\top$—exhibit clearly distinguishable characteristics. Furthermore, we plotted the distribution of scores across different model blocks, as shown in Fig. 5. It can be observed that the values of $Q^{\text{src}}(K^{\text{src}})^\top$ are significantly larger than those of $Q^{\text{src}}(K^{\text{ctx}})^\top$, and this trend becomes more pronounced in deeper layers.

**Implementation of *Pre-Selection*.** Given that context tokens exhibit significantly lower saliency than source tokens within the attention mechanism, we posit that the majority of context tokens are redundant. By leveraging pooling attention, we derive the compressed score matrix

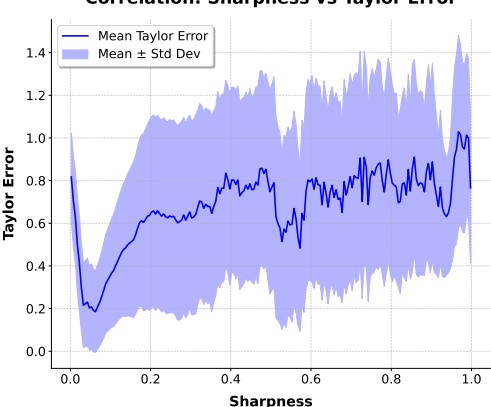

*Figure 6.* This visualization results demonstrate that the Taylor error $\mathcal{E}_i$ is directly proportional to the coarse-grained variance $M_i$ (sharpness). Consequently, $M_i$ serves as an effective indicator of the magnitude of $\mathcal{E}_i$, providing empirical support for the conclusion of Theorem 3.1. The distribution of $||Q(K-K^c)^\top||_\infty^2$ is in Fig. 16.

$S_{\text{coarse}} \in \mathbb{R}^{B \times H \times N_Q \times N_K}$. Let $L_{src}$ and $L_{ctx}$ denote the number of source and context tokens, respectively, satisfying $L_{src} + L_{ctx} = N = S$ (where we assume divisibility by the block size $L_Q$ for notational simplicity). Consequently, the slice $S_{\text{coarse}}[:,:,:\lceil L_{src}/L_Q\rceil,:\lceil L_{src}/L_Q\rceil]$ corresponds to the scores of the source tokens, whereas $S_{\text{coarse}}^{\text{ctx}} = S_{\text{coarse}}[:,:,:\lceil L_{src}/L_Q\rceil, \lceil L_{src}/L_Q\rceil:]$ represents those of the context tokens. Therefore, the importance ranking of context Key/Value pairs is as $I_{\text{topk}} \in \mathbb{R}^{B \times H \times N_K}$

$$= \text{TopK}(\text{Mean}(S_{\text{coarse}}^{\text{ctx}}, \text{axis} = 2), \text{axis} = 2). \quad (1)$$

Subsequently, we finalize *pre-selection* by reconstructing the sparse tensors via *gather* and *concatenation* operations:

$$K_{\text{sel}}^{\text{ctx}} = \text{Gather}(K^{\text{ctx}}, I_{\text{topk}}), V_{\text{sel}}^{\text{ctx}} = \text{Gather}(V^{\text{ctx}}, I_{\text{topk}})$$
$$K_{\text{new}} = \text{Concat}([K^{\text{src}}, K_{\text{sel}}^{\text{ctx}}]), V_{\text{new}} = \text{Concat}([V^{\text{src}}, V_{\text{sel}}^{\text{ctx}}]). \quad (2)$$

We introduce a hyperparameter, the Select Ratio $\alpha_s$, which dictates that the Top-$k$ operator retains the $\alpha_s \lceil L_{\text{ctx}}/b\rceil$ most salient context blocks. This mechanism effectively reduces the computational complexity from $\mathcal{O}(NSD)$ to $\mathcal{O}(N(L_{\text{src}} + \alpha_s L_{\text{ctx}})D)$.

**Block-Wise 0-th order Taylor Sparse Attention.** Upon deriving the compressed tensors $K_{\text{new}}$ and $V_{\text{new}}$, we partition the Queries and route them to distinct computational kernels: standard FlashAttention v2/3 and sparse attention. Specifically, the sparse attention denotes *block-wise 0-th order Taylor sparse attention*. Within this mechanism, the fundamental strategy is to select a subset of salient blocks for exact computation via OnlineSoftmax($Q_i K_j^\top)V_j$, while the remaining blocks utilize a 0-th order Taylor approximation of the Key and Value tensors to accelerate processing. Here, OnlineSoftmax denotes the block-wise computation of the softmax function (refer to Algorithm 2, lines 19–26). As illustrated in the corresponding part of Fig. 3, the

implementation of this sparse attention mechanism necessitates the pre-computation of the compressed tensors $K^c$ and $V^c$, alongside the pooling score matrix $S_{\text{coarse}}$ and the block mask $M_{\text{coarse}}$. For a given Query block $Q_i \in \mathbb{R}^{B \times H \times L_Q \times D}$, the computational pathway is determined conditionally by the mask entries $M_{ij}$. Specifically, when $M_{ij} = 1$, the following operation is executed:

$$S_{ij} = Q_i K_j^\top \cdot 1/\sqrt{D}, \ P_{ij} = \exp(S_{ij}), \tag{3}$$
$$\ell_i += \text{rowsum}(P_{ij}), \ O_i += P_{ij} V_j,$$

where $\ell_i$ and $O_i$ denote the softmax normalization factor and the output accumulator, respectively. For the sake of clarity, we omit the subtraction of the maximum value typically applied for numerical stability in the exponential calculation.

Conversely, when $M_{ij} = 0$, the block interaction term $\exp(Q_i K_j^\top)V_j$ is approximated via its 0-th order Taylor expansion as $\exp(Q_i (K_j^c)^\top)V_j^c$. This approximation reduces the computational complexity from $\mathcal{O}(L_Q L_K D)$ to $\mathcal{O}(L_Q D)$, and the update rule is formulated as follows:

$$S_{ij}^c = Q_i (K_j^c)^\top \cdot 1/\sqrt{D}, \ P_{ij}^c = \exp(S_{ij}^c), \tag{4}$$
$$\ell_i += P_{ij}^c \cdot L_K, \ O_i += P_{ij}^c V_j^c \cdot L_K.$$

Finally, the calculation is completed by performing $O_i = O_i/\ell_i$. We also investigated 1st- and 2nd-order Taylor expansions. However, implementation trials revealed that these variants are ill-suited for hardware acceleration and incur prohibitive computational overhead. Consequently, they were discarded from the final design. A comprehensive algorithm flowchart is provided in Appendix G.

**Theorem 3.1.** *(Proof in Appendix F) Let $S_1(i) = \sigma(z_i)$ and $S_2(i) = \sigma(\tilde{z}_i)$ be the true and approximate attention distributions for token $i \in Block_u$. Let $M_u \triangleq \mathbb{E}_{i \in Block_u}[\|S_2(i)\|_2^2]$ be the **block-wise sharpness metric**. Assuming the attention energy $f(i) = \|S_2(i)\|_2^2$ is L-Lipschitz within the block, where $L$ is determined by the spectral norms of projection weights $\|W_Q\|, \|W_K\|$, the expected approximation error $\mathcal{E}_u = \mathbb{E}_{i \in Block_u}[\|S_1(i) - S_2(i)\|_2^2]$ is bounded by:*

$$\mathcal{E}_u \leq (M_u + L\delta) \cdot \mathbb{E}_{i \in Block_u}[\|\Delta z_i\|_2^2] + \mathcal{C}_H \mathbb{E}[\|\Delta z_i\|_2^4] \tag{5}$$

*where $\delta$ is the block diameter and $\mathcal{C}_H$ is a constant related to the maximum curvature (Hessian) of the softmax function.*

***Grouped Computation.*** Intuitively, Queries can be dichotomized based on the magnitude of the approximation error induced by sparse attention: those exhibiting high error and those with negligible error. The proof of Theorem 1 establishes that the error of the block-wise 0-th order Taylor sparse attention is upper-bounded by the product of $\|Q(K - K^c)^\top\|_\infty^2$ and $M_i = [\text{Var}(\text{softmax}(Q_i^c(K^c)^\top))]$. Here, the former term characterizes the mean intra-block

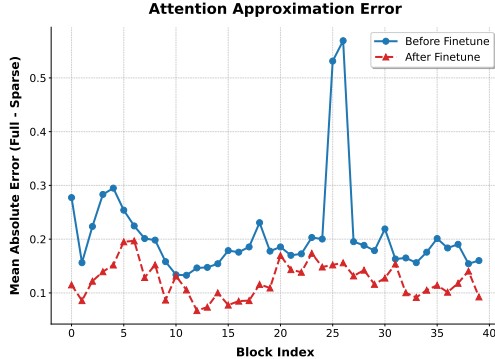

*Figure 7.* ISA is a trainable sparse attention mechanism. After post-training, the discrepancy between the output of ISA and that of full attention is significantly reduced across nearly all blocks.

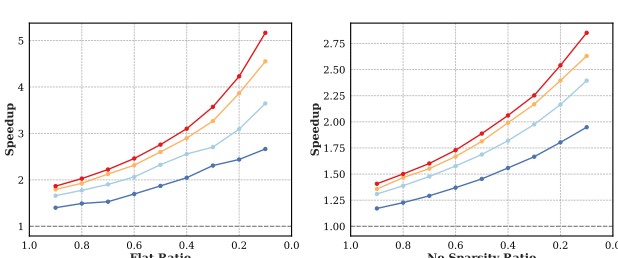

*Figure 8.* Reducing the Flat Ratio $\alpha_f$ and No Sparsity Ratio $\alpha_{ns}$ leads to an exponential increase in the speedup of ISA relative to SDPA. Moreover, this speedup becomes increasingly pronounced as sequence length grows.

variance, while the latter quantifies the variance between block means. This suggests that both $\|Q(K - K^c)^\top\|_\infty^2$ and $M_i$ are potential candidates for indexing the Taylor approximation error, thereby facilitating the grouping of Queries. However, computing the former is computationally prohibitive, and we further demonstrate in Appendix D.1 that it is an ineffective proxy for the Taylor error. In contrast, $M_i$, derived efficiently from the pooling score matrix, exhibits a strong positive correlation with the Taylor error, as evidenced in Fig. 6. Consequently, we adopt $M_i$ as our selection metric, formally defining it as sharpness. Under this adaptive framework, Queries exhibiting high sharpness (indicating high Taylor error) are routed to the standard attention branch, whereas those with low sharpness (indicating low Taylor error) are processed via block-wise 0-th order Taylor sparse attention. This strategy effectively mitigates the adverse impact of high-error Queries, enabling ISA to achieve a sparsity of 93.75% in the Taylor sparse attention component with negligible performance degradation.

**Implementation Detail and Analysis. First**, we implemented the forward pass of the block-wise 0-th order Taylor sparse attention using both Triton (Tillet et al., 2019) and TileLang (Wang et al., 2025). A comprehensive performance comparison between the two implementations is provided in Appendix H. Furthermore, we developed the backward pass using Triton to establish ISA as a fully dif-

*Table 1.* **EditVerseBench Evaluation Results.** Our proposed **LIVEditor** (ISA) outperforms related methods across nearly all metrics and surpasses **LIVEditor** (full-attn).

| Method | VLM Evaluation | | | | Pick Score | |
|---|---|---|---|---|---|---|
| | Quality | Text Align | Temporal Consistency | Editing Quality | Frame | Video |
| **Attention Manipulation (Training-free)** | | | | | | |
| TokenFlow | 6.09 | 19.97 | 26.64 | 24.05 | 98.67 | 98.82 |
| STDF | 5.70 | 19.70 | 26.34 | 23.43 | 96.31 | 96.08 |
| **First-Frame Propagation (w/ End-to-End Training)** | | | | | | |
| Señorita-2M | 7.54 | 19.71 | 27.14 | 24.32 | 98.03 | 98.12 |
| **Instruction-Guided (w/ End-to-End Training)** | | | | | | |
| InsV2V | 5.95 | 19.52 | 25.67 | 22.77 | 97.22 | 96.83 |
| Lucy Edit | 6.70 | 19.52 | 26.23 | 23.28 | 98.56 | 98.44 |
| EditVerse | 7.65 | 20.07 | 26.73 | 23.93 | 98.56 | 98.42 |
| **LIVEditor** (full-attention) | 7.62 | 19.98 | 27.13 | 23.80 | 99.24 | 99.19 |
| **LIVEditor** (ISA) | **7.89** | **20.09** | **27.19** | **24.55** | **99.32** | **99.22** |

*Table 2.* **EditVerseBench Evaluation Results.** In the training-free scenarios, we compare our proposed ISA against several attention mechanism variants, including Radial, Sparse, STA, SWA, and VSA. ISA significantly outperforms these sparse baselines across all metrics. Furthermore, ISA surpasses full attention on all metrics, with the exception of Pick Score (Video).

| Method | VLM Evaluation | | | | Pick Score | | SpeedUp |
|---|---|---|---|---|---|---|---|
| | Quality | Text Align | Temporal Consistency | Editing Quality | Frame | Video | Compare to FA3 |
| Radial Attention | 7.09 | 19.68 | 26.86 | 24.13 | 98.44 | 98.92 | 1.28 |
| Sparge Attention | 7.44 | 19.69 | 26.75 | 23.76 | 98.64 | 98.98 | 1.40 |
| STA | 4.45 | 15.76 | 13.02 | 4.82 | 94.06 | 93.65 | **2.09** |
| SWA | 5.95 | 18.48 | 20.06 | 16.74 | 97.80 | 98.13 | 1.37 |
| VSA | 3.60 | 16.85 | 17.30 | 9.88 | 94.99 | 94.19 | 1.38 |
| **LIVEditor** (full-attention) | 7.62 | 19.98 | 27.13 | 23.80 | 99.24 | **99.19** | 1.00 |
| **LIVEditor** (ISA) | **7.78** | **20.07** | **27.15** | **24.15** | **99.26** | 99.15 | 1.47 |

ferentiable and trainable sparse attention mechanism. As demonstrated in Fig. 7, fine-tuning ISA significantly mitigates the approximation error relative to the standard attention baseline. **Second**, the sparsity profile of ISA is governed by three hyperparameters: the Select Ratio $\alpha_s$, the No-Sparsity Ratio $\alpha_{ns}$, and the Flat Ratio $\alpha_f$. These parameters respectively determine the fraction of context tokens retained during pre-selection, the density of the Taylor sparse attention, and the proportion of Queries routed to the standard attention branch during grouped computation. Crucially, a reduction in the values of these parameters corresponds to an increase in the overall sparsity of the ISA mechanism. Fig. 8 demonstrates that the computational speedup achieved by ISA increases monotonically as the values of $\alpha_f$ and $\alpha_{ns}$ are reduced.

**LIVEditor-14B.** Building upon the efficiency of ISA, we introduce LIVEditor-14B, a unified framework for lightning-fast video editing. To maximize editing robustness and fidelity, LIVEditor-14B incorporates three critical design choices. **First**, it seamlessly integrates ISA as the core attention mechanism, enabling the processing of long ICL sequences with negligible computational overhead. **Second**, we adopt a progressive two-stage training paradigm to balance generalization and precision. The model is first pre-trained on a large-scale, mixed-quality dataset (1.7M samples) to learn broad editing semantics, followed by fine-tuning on a highly curated subset of high-quality data (0.089M samples) to refine visual aesthetics

and instruction adherence. **Third**, to address the length discrepancy between source and context videos inherent in editing tasks, we introduce a decoupled Rotary Positional Embedding (RoPE) strategy. Specifically, we apply RoPE independently to source and context tokens, resetting positional indices to zero for each group, thereby preventing positional bias and ensuring robust performance across variable sequence lengths.

## 4. Experiment

### 4.1. Setup

**Data Pipeline.** Our dataset is derived from two primary sources: *self-constructed data* and *publicly available datasets*. For the self-constructed portion, we first employ *Gemini 2.5 Flash* (Comanici et al., 2025) to generate descriptive captions for the original videos. We then prompt *Gemini 2.5 Flash* to select a specific editing subtask, such as object addition, removal, swapping, background alteration, or style transfer, and synthesize modification instructions for the initial frame. Subsequently, we utilize *Gemini 2.5 Image Preview* (Google, 2025a) to generate the corresponding edited image. To maintain temporal consistency, we apply pose guidance for human-centric videos via text-and-image-to-video (TI2V) while attention injection is used for non-human subjects. However, given that this method exhibited suboptimal consistency for non-human subjects, we augmented our training data with public datasets including

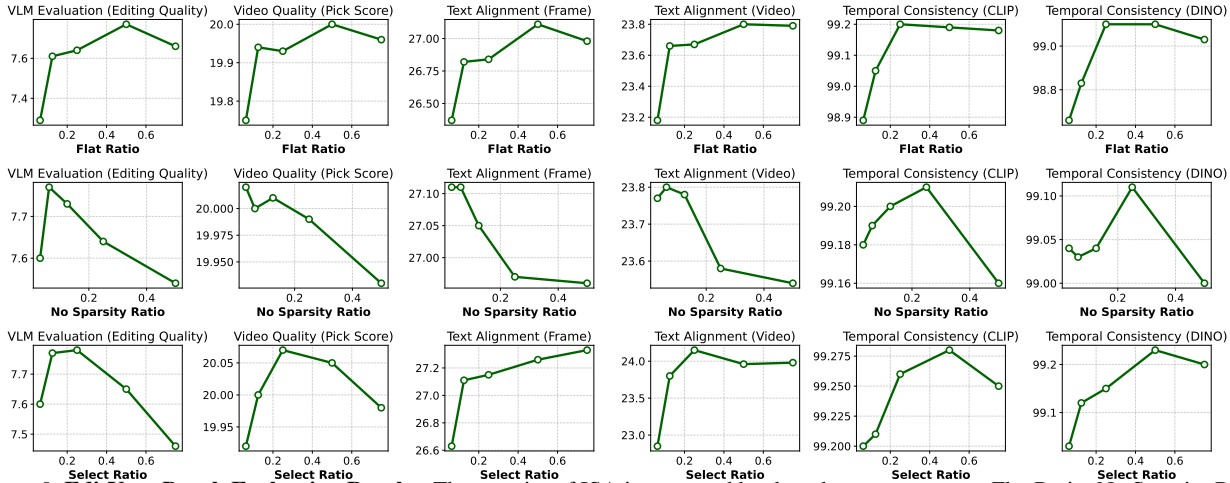

*Figure 9.* **EditVerseBench Evaluation Results.** The sparsity of ISA is governed by three hyperparameters: Flat Ratio, No Sparsity Ratio, and Select Ratio. Lower values for these parameters induce greater sparsity. Our ablation studies indicate that ISA is particularly sensitive to the Flat Ratio, with performance degrading significantly as this value decreases. In contrast, the model exhibits robustness to variations in the No Sparsity Ratio and Select Ratio. Consequently, we maintain a relatively high Flat Ratio (defaulting to 0.5) in our experiments, while setting the No Sparsity Ratio and Select Ratio to lower values (0.0625 and 0.125, respectively).

Ditto (Bai et al., 2025), LoVoRA (Xiao et al., 2025), and ReCo (Zhang et al., 2025c). A detailed analysis of the data distribution and the data pipeline is provided in Appendix C.

**Benchmarks.** We evaluate our method on four benchmarks: EditVerseBench (Ju et al., 2025), VIE-Bench (Mou et al., 2025), IVE-Bench (Chen et al., 2025b), and FiVE-Bench (Li et al., 2025a). Specifically, we use Edit-VerseBench, VIE-Bench, and IVE-Bench to validate the superiority of LIVEditor-14B over existing methods. We further employ EditVerseBench to benchmark ISA against other sparse attention mechanisms. Finally, ablations on EditVerseBench and FiVE-Bench confirm that ISA outperforms full attention and analyze the impact of hyperparameters $\alpha_s, \alpha_{ns},$ and $\alpha_f$. Appendix A provides detailed benchmark descriptions.

**Model.** We derive our video editing model via post-training on the high-noise branch of Wan 2.2. The training regimen proceeds in two distinct stages. The first stage utilizes 1.7M samples with a learning rate of $1e^{-5}$ and a global batch size of 16. The second stage employs 0.089M high-quality samples with a reduced learning rate of $1e^{-6}$ and a global batch size of 16. Both stages utilize the DeepSpeed ZeRO-3 Offload optimization strategy (Microsoft, 2022). Furthermore, to mitigate artifacts from unrealistic synthetic data, we exclusively employ a configuration where synthetic images serve as context tokens while real images function as source tokens. Finally, we set the default values of $\alpha_s$, $\alpha_{ns}$, and $\alpha_f$ to 0.125, 0.0625, and 0.5 respectively. Comprehensive hyperparameter configurations are provided in Appendix B.

### 4.2. Main Result

**Evaluation on EditVerseBench.** As illustrated in Table 1, we evaluate LIVEditor-14B trained with ISA, denoted as LIVEditor-14B (ISA), and LIVEditor-14B trained with full attention, denoted as LIVEditor-14B (full-attn), on EditVerseBench. We compare them against state-of-the-art methods including TokenFlow (Qu et al., 2025), STDF (Gao et al., 2025), Señorita-2M (Zi et al., 2025b), InsV2V (Cheng et al., 2023), Lucy Edit (Team, 2025) and EditVerse (Ju et al., 2025). Experimental results demonstrate that both LIVEditor-14B (ISA) and LIVEditor-14B (full-attn) achieve leading performance across all metrics. Specifically, in VLM evaluations, LIVEditor-14B (ISA) obtains scores of 7.89, 20.09, 27.19, and 24.55 for Quality, Text Alignment, Temporal Consistency, and Editing Quality, respectively. These scores surpass the previous best performances of 7.65, 20.07, 27.14, and 24.32 by margins of 0.24, 0.02, 0.05, and 0.23. Regarding Pick Scores, LIVEditor-14B (ISA) reaches 99.32 on frames and 99.22 on video. These results exceed the previous highest records of 98.56 and 98.44 by 0.76 and 0.78. Finally, LIVEditor-14B (ISA) outperforms LIVEditor-14B (full-attn) on all metrics with the exception of the Video Pick Score.

**Evaluation on Other Benchmarks.** We further evaluate LIVEditor-14B (ISA) and LIVEditor-14B (full-attn) on VIE-Bench and IVE-Bench against an expanded set of state-of-the-art video editing methods. In addition to the previously mentioned baselines, we include comparisons with Ditto (Bai et al., 2025), VACE (Jiang et al., 2025), ICVE (Liao et al., 2025), Omni-Video (Liang et al., 2025), AnyV2V (Ku et al., 2024), StableV2V (Liu et al., 2025a), and Pika (PiKa, 2025). Detailed results are pro-

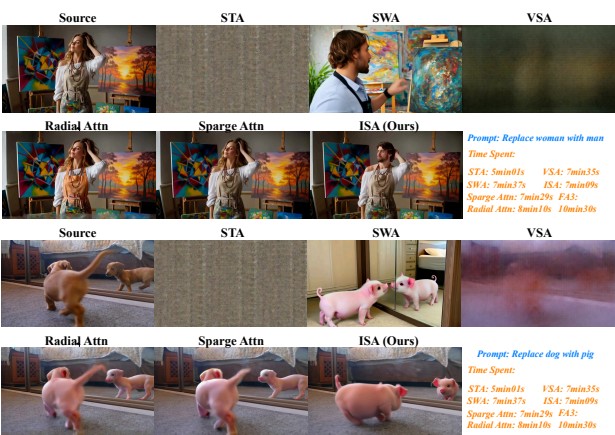

*Figure 10.* **ISA Performance Visualization.** Our proposed ISA not only surpasses VSA, SWA, Sparge Attn, and Radial Attn in inference speed but also outperforms all other sparse attention mechanisms in terms of model performance.

*Table 3.* **EditVerseBench Evaluation Results.** The model performance is further enhanced following fine-tuning with the high-quality dataset in Stage II.

| Method | VLM Evaluation | | | |
|---|---|---|---|---|
| | Quality | Text Align | Temporal Consistency | Editing Quality |
| **LIVEditor** (ISA, Stage I) | 6.46 | 19.50 | 25.27 | 22.63 |
| **LIVEditor** (ISA, Stage II) | 7.89 | 20.09 | 27.19 | 24.55 |

vided in Appendix E due to space constraints. These experiments demonstrate that `LIVEditor-14B` (ISA) achieves the best overall performance.

**Sparse Attention Comparison.** To demonstrate the superiority of ISA in video editing, we benchmark it against prominent sparse attention algorithms on EditVerseBench. These baselines include Radial Attention (Zhang et al., 2024), Sparge Attention (Zhang et al.), STA (Zhang et al., 2025b), SWA (Fu et al., 2025), and VSA (Zhang et al., 2025a). We present the detailed results in Table 2. For a fair comparison, we apply ISA directly to the pre-trained `LIVEditor-14B` (full-attn) in a training-free manner. The results indicate that `LIVEditor-14B` (ISA) significantly outperforms all other sparse attention methods. Notably, it even surpasses the original `LIVEditor-14B` (full-attn). We hypothesize that this improvement stems from the ability of ISA to discard irrelevant noise tokens. Furthermore, while ISA is marginally slower than STA, the latter suffers from severe performance degradation. Finally, we visualize the qualitative effects of different sparse attention mechanisms in Fig. 10. We observe that Radial Attention and Sparge Attention perform relatively well yet fail to fully adhere to the editing instructions. In contrast, ISA demonstrates strong temporal consistency executes the requested modifications.

### 4.3. Ablation Studies

**Stage I vs. Stage II.** Our `LIVEditor-14B` follows a two-stage training paradigm. The first stage utilizes a mixture of high-quality and low-quality data. The second stage refines the model using a high-quality subset integrated from our

*Table 4.* **VIE-Bench Evaluation Results.** The model performance is further enhanced following fine-tuning with the high-quality dataset in Stage II.

| Method | Model Type | | VIE-Bench Score | | | |
|---|---|---|---|---|---|---|
| | Comm. | Base | Follow↑ | Pres.↑ | Qual.↑ | Avg.↑ |
| **LIVEditor** (ISA, Stage I) | ✓ | ✓ | 1.31 | 8.31 | 5.87 | 5.16 |
| **LIVEditor** (ISA, Stage II) | ✓ | ✓ | **5.55** | **8.57** | **6.33** | **6.81** |

constructed dataset and open-source collections. To validate the importance of this second stage, we evaluate the model on EditVerseBench and present the results in Table 3 and Table 4. The outcomes demonstrate that the Stage II model yields improved performance across all metrics.

**Hyperparameters in ISA.** The sparsity of ISA is governed by three hyperparameters: the Flat Ratio $\alpha_f$, the No Sparsity Ratio $\alpha_{ns}$, and the Select Ratio $\alpha_s$. To analyze their impact, we visualize performance trends in Fig. 9 by varying one hyperparameter while keeping the others fixed. The default values are set to $\alpha_f = 0.5$, $\alpha_{ns} = 0.0625$, and $\alpha_s = 0.125$. We conduct this evaluation in a training-free manner by applying ISA directly to the pre-trained full-attention model. As for $\alpha_f$, we observe high sensitivity. Performance declines significantly across all metrics as sparsity increases. Consequently, we maintain $\alpha_f$ at 0.5 for our trainable models. For $\alpha_{ns}$, the Video Quality (Pick Score) and Text Alignment metrics improve with increasing sparsity. Other metrics exhibit an initial rise followed by a decline. This suggests that $\alpha_{ns}$ can be set to a low value without compromising performance. Thus, we default it to 0.0625. Finally, $\alpha_s$ displays an initial increase followed by a decrease on most metrics. This indicates that a small value is also sufficient. However, the optimal $\alpha_s$ is slightly larger than the optimal $\alpha_{ns}$. Therefore, we select a default value of 0.125. We employ this parameter set as the default for all comparative experiments. Furthermore, as shown in Table 2, this configuration achieves a significant 1.47x speedup compared to full attention in the end-to-end scenario.

## 5. Conclusion

In this work, we presented `LIVEditor-14B`, a lightning and unified framework for video editing. We addressed the computational bottleneck of ICL by introducing **In-context Sparse Attention** (**ISA**). This novel mechanism is grounded in our theoretical analysis linking query sharpness to Taylor approximation errors. By dynamically routing Queries and pruning redundant context tokens, ISA achieves experimentally lossless acceleration. Furthermore, we established a comprehensive data processing pipeline to curate a high-quality video editing dataset. Extensive evaluations on EditVerseBench and other benchmarks demonstrate that `LIVEditor-14B` significantly outperforms SOTA methods in both generation quality and inference latency. We believe that the principles underlying ISA offer a scalable solution for future long-sequence video generation tasks.

## Impact Statement

We present **LIVEditor-14B**, a framework designed for efficient and unified video editing via ISA. The dataset utilized by **LIVEditor-14B** is curated from a combination of self-constructed samples generated by foundation models and publicly available academic datasets (e.g., Ditto, LoVoRA). To ensure data safety, this dataset undergoes rigorous filtering through automated safety assessment pipelines to guarantee compliance with ethical standards and the exclusion of harmful content. Furthermore, given the generative nature of video editing, we place a strong emphasis on the responsible and ethical deployment of this technology. We strive to maximize its societal benefits in creative industries while diligently addressing and mitigating potential risks associated with misuse, such as the creation of misleading content.

## Acknowledgments

This work was supported by the National Natural Science Foundation of China under Grant No. 62506317 and Guangdong Provincial Key Lab of Integrated Communication, Sensing and Computation for Ubiquitous Internet of Things (No.2023B1212010007). This work was partially supported by The Hong Kong University of Science and Technology (Guangzhou) Kunpeng&Ascend Center of Cultivation.

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

# A. Evaluation Benchmark

To comprehensively evaluate the effectiveness, robustness, and generalization capabilities of our proposed method, we conduct extensive experiments across 4 representative and challenging benchmarks: EditVerseBench (Ju et al., 2025), IVE-Bench (Chen et al., 2025b), VIE-Bench (Mou et al., 2025), and FiVE-Bench (Li et al., 2025a). By leveraging this multifaceted evaluation framework, we aim to provide a rigorous assessment of the our method's performance under varying degrees of complexity and diverse editing requirements. The specific characteristics and composition of each benchmark are detailed as follows:

**EditVerseBench (Ju et al., 2025).** EditVerseBench is a comprehensive benchmark designed to address the shortcomings of previous video editing datasets. The benchmark is meticulously curated from 100 high-quality, real-world videos sourced from Pixabay, encompassing a wide variety of scenes. A key feature of its design is the inclusion of both 50 horizontal and 50 vertical videos, reflecting contemporary video formats. Each video is paired with two distinct editing instructions, resulting in a total of 200 video-instruction pairs. These pairs are evenly distributed across 20 distinct editing categories, which span a broad spectrum of tasks, including object manipulation, stylization, background and weather changes, inpainting, and reasoning-based edits. This diverse and balanced composition enables a rigorous assessment of a model's ability to adhere to complex instructions, maintain visual quality, and preserve temporal consistency across various editing scenarios and aspect ratios.

**IVE-Bench (Chen et al., 2025b).** IVE-Bench is designed to overcome the limitations of limited source diversity and narrow task coverage in existing benchmarks. It features a robust corpus of 600 high-quality source videos selected to span 7 semantic dimensions, including Subject, Theme, Time, Perspective, Motion, Emotion, and Scene. A distinguishing characteristic of IVE-Bench is its focus on temporal scalability. The dataset is partitioned into a short subset (400 videos, 32-128 frames) and a long subset (200 videos, 129-1,024 frames), allowing for a rigorous assessment of long-sequence consistency. However, in our experiments, due to the resource constraints, we only conduct the evaluations on short subsets. The benchmark defines 8 major editing categories subdivided into 35 fine-grained subcategories, totaling 600 LLM-refined instruction prompts. Furthermore, IVE-Bench establishes a 3-dimensional evaluation protocol that assesses Video Quality, Instruction Compliance, and Video Fidelity using both traditional metrics and MLLMs to ensure high alignment with human perception.

**VIE-Bench (Mou et al., 2025).** VIE-Bench is a high-quality benchmark developed to address the scarcity of instruction-based video editing datasets that support complex and reference-guided tasks. The benchmark comprises 140 high-quality video instances, sourced from public datasets such as DAVIS and HumanVid as well as the web. All videos are curated at 720p resolution with durations ranging from 3 to 10 seconds, covering a wide array of scenes including indoor, outdoor, dynamic, animated, and portrait environments. The editing tasks are categorized into 3 main types with 8 fine-grained sub-taks: Local Editing (Add, Remove, Object Swap, Color Change), Global Editing (Style Change, Tone/Weather Change), and Reference-Based Editing (Reference Base Swap, Reference Base Add). In our experiments, we do not cover the samples in Reference-Based Editing. To ensure robust evaluation, VIE-Bench abandons traditional CLIP-based metrics in favor of an MLLM-based evaluation protocol, utilizing GPT-4o as a judge to score results on Instruction Following, Preservation (temporal and semantic consistency), Quality (visual aesthetics), and Similarity (for reference-based tasks), ensuring alignment with human perception.

**FiVE-Bench (Li et al., 2025a).** FiVE-Bench is utilized to evaluate the model's precision in fine-grained, object-level video editing. Unlike benchmarks that focus on holistic style transfer, FiVE-Bench is specifically designed to assess precise modifications while preserving the original context. The benchmark comprises 100 videos, including 74 real-world clips meticulously curated from the DAVIS dataset and 26 high-quality synthetic videos, ensuring a diverse range of motion and complexity. These videos are paired with 420 source-target prompt pairs and corresponding segmentation masks annotated via SAM2. The benchmark categorizes tasks into 6 fine-grained editing types of varying difficulty: color alteration, material modification, object substitution (distinguishing between rigid and non-rigid deformations), object addition, and object removal. To overcome the limitations of traditional CLIP scores in detecting subtle object changes, FiVE-Bench introduces FiVE-Acc, a novel evaluation metric that leverages VLMs to verify editing success through semantic Yes/No and multiple-choice questions, enabling a rigorous assessment of semantic fidelity and instruction adherence.

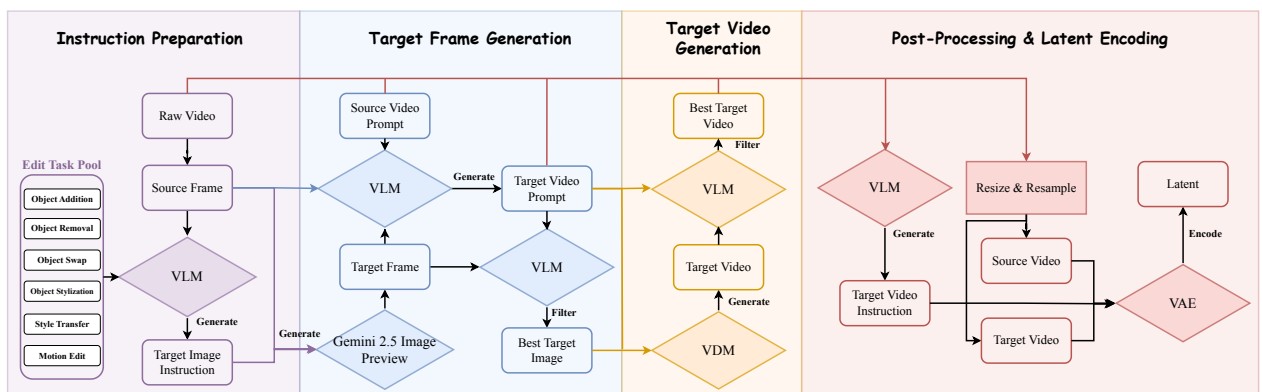

*Figure 11.* **The automated synthesis pipeline for video-to-video editing**. Our framework consists of 4 integrated stages: **(1) Instruction Preparation,** where a VLM samples tasks from the Edit Task Pool to generate precise target image instructions based on raw video frames; **(2) Target Frame Generation,** utilizing the Gemini 2.5 Image Preview to synthesize an anchor frame followed by a VLM-based filtering loop; **(3) Target Video Generation**, where a VDM produces the final video sequence subject to further quality-aware VLM selection; and **(4) Post-processing & Latent Encoding,** involving temporal resampling, instruction annotation, and VAE-based encoding into the latent space for downstream training.

*Table 5.* **FiVE-Bench Results (Object Replacement (Rigid)).** **LIVEditor** (ISA) outperforms **LIVEditor** (full-attn) on almost all metrics.

| Method | Structure | Background Preservation | | | | Text Align | IQA | | Temp. Consis. |
|---|---|---|---|---|---|---|---|---|---|
| | Dist.↓ | PSNR↑ | LPIPS↓ | MSE↓ | SSIM↑ | CLIP-S↑ | CLIP-S↑ | NIQE↓ | Motion Fid.↑ |
| **LIVEditor (full-attention)** | 81.31 | 14.52 | 368.36 | 519.53 | 47.38 | 25.42 | 20.13 | **6.04** | 55.73 |
| **LIVEditor (ISA)** | **71.37** | **15.40** | **310.87** | **422.79** | **52.31** | **26.67** | **20.76** | 6.30 | **58.48** |

## B. Hyperparameter Setting

We initialize our framework using the high-noise variant of the pre-trained Wan2.2-T2V model. Training is conducted in two stages using the AdamW optimizer with $\beta_1 = 0.9$, $\beta_2 = 0.999$, and a weight decay of $10^{-2}$. In the first stage, we train on approximately 1.7 million samples with a learning rate of $10^{-5}$. We utilize 32 80GB GPUs distributed across 4 nodes, employing sequence parallelism of size 2 to maintain a global batch size of 16. To facilitate ICL, we introduce a specialized RoPE strategy. Specifically, we apply RoPE independently to source and context tokens, resetting positional indices to zero for each, which mitigates robustness issues caused by varying lengths between source and target videos during inference. In the second stage, we perform fine-tuning on 0.089 million samples to optimize performance. During this phase, the learning rate is reduced to $5 \times 10^{-6}$, and we utilize Exponential Moving Average (EMA) with a decay rate of 0.995 to derive the final video editing model.

In all comparative experiments, we implement ISA using a Triton-based Block-Wise 0-th Taylor Sparse Attention mechanism. Regarding hyperparameters, we adopt a flat ratio of 0.5, a no-sparsity ratio of 0.0625, and a select ratio of 0.125. Under these settings, ISA achieves a $1.47\times$ speedup over FlashAttention-3 during full 32-step inference (with CFG), while simultaneously demonstrating superior performance.

For the remaining sparse attention baselines, we adopt the following configurations: VSA and Sparge Attention are set to sparsity rates of 87.5% and 82.5%, respectively. STA utilizes the official kernel implementation with a window size of $9 \times 9 \times 9$. Radial Attention employs the default radial mask and performs inference using SageAttention-2. Finally, SWA uses a default window size of $2040 \times 5$ (comprising five latent vectors of size 2040), processing a total of $2040 \times 30$ tokens during inference.

## C. Data Pipeline

This section details the data processing pipeline employed in **LIVEditor-14B**, analyzes the data distribution, and describes the scheduling strategies utilized during training.

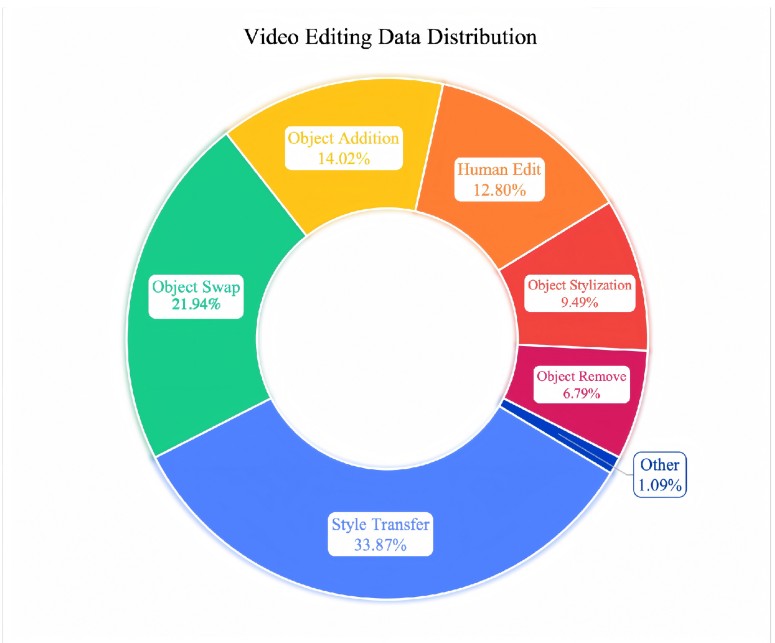

*Figure 12.* **Distribution of Editing Tasks in the Constructed Dataset.** The dataset comprises diverse editing scenarios, categorized into seven primary tasks. Global editing (e.g., Style Transfer) constitutes the largest portion to ensure stylistic diversity, while fine-grained semantic edits (e.g., Object Swap, Addition, Human Edit) are heavily represented to enhance the model's instruction-following precision on localized regions.

*Table 6.* **FiveBench Results (Object Replacement (Rigid)). `LIVEditor`** (ISA) outperforms **`LIVEditor`** (full-attn) on almost all metrics.

| Method | FiVE Evaluation | | | | |
|---|---|---|---|---|---|
| | YN↑ | MC↑ | FiVE-∪↑ | FiVE-∩↑ | Acc.↑ |
| **`LIVEditor` (full-attention)** | 52.00 | 53.00 | 25.86 | 44.33 | 43.80 |
| **`LIVEditor` (ISA)** | **59.00** | **58.00** | **37.88** | **47.94** | **50.71** |

### C.1. Data Processing Pipeline

The proposed video-to-video data synthesis pipeline, illustrated in Fig. 11, facilitates the automated generation of high quality, instruction-aligned video editing pairs. The workflow is systematically partitioned into 4 primary phases:

**Instruction Preparation.** The process begins with the ingestion of raw video data, from which a representative source frame is extracted. In our pipeline, we usually extract the first frame of the raw video data as the source frame. Then we utilize the VLMs (Gemini-2.5-Pro/GPT-4o) to randomly sample from an edit task pool, which includes object addition, object removal, object swap, object stylization, style transfer, and motion edit, as the editing task type to generate a precise target image instruction based on the content of the input source frame.

**Target Frame Generation.** After we get the target image instruction, we utilize the Gemini 2.5 Image Preview (Gemini-2.5-Image-Preview) to transform the source frame into a target frame conditioned on the synthesized instruction. In the next step, we input the source frame, the source video prompt, and the target frame into the VLMs (Gemini-2.5-Pro/GPT-4o) to generate the target video prompt, one system prompt we used is shown in Fig. 13. In order to keep the quality of the data, we utilize the VLMS (Gemini-2.5-Pro/GPT-4o), HPSv2 (Wu et al., 2023b), PickScore (Kirstain et al., 2023) and other metrics to filter the generated target frame, along with the inputs of target frame, target video prompt and the target image instruction.

**Target Vide Generation.** In this stage, we utilize our internal 14-billion-parameter text-to-image-to-video diffusion model to synthesize the target video based on the validated target image and augmented text prompts. To ensure consistency between the source video and the generated target video, we first compute inconsistency scores using YOLO (Khanam & Hussain, 2024), GroundingDINO (Ren et al., 2024), and SAM (Kirillov et al., 2023) to filter out low-quality samples. Subsequently,

*Table 7.* **FiVE-Bench Results (Object Replacement (Non-Rigid)).** `LIVEditor` (ISA) outperforms `LIVEditor` (full-attn) on almost all metrics.

| Method | Structure | Background Preservation | | | | Text Align | IQA | | Temp. Consis. |
|---|---|---|---|---|---|---|---|---|---|
| | Dist. $(\times 10^3)\downarrow$ | PSNR$\uparrow$ | LPIPS $(\times 10^3)\downarrow$ | MSE $(\times 10^4)\downarrow$ | SSIM $(\times 10^2)\uparrow$ | CLIP-S$\uparrow$ | CLIP-S (edit)$\uparrow$ | NIQE$\downarrow$ | Motion Fid. $(\times 10^2)\uparrow$ |
| `LIVEditor` (full-attention) | 90.55 | 14.09 | 359.51 | 540.00 | 47.52 | **24.72** | **19.16** | **5.93** | 54.59 |
| `LIVEditor` (ISA) | **81.78** | **14.75** | **319.22** | **461.22** | **51.21** | 24.68 | 19.07 | 6.29 | **57.99** |

*Table 8.* **FiVE-Bench Results (Object Replacement (Non-Rigid)).** `LIVEditor` (ISA) outperforms `LIVEditor` (full-attn) on almost all metrics.

| Method | FiVE Evaluation | | | | |
|---|---|---|---|---|---|
| | YN$\uparrow$ | MC$\uparrow$ | FiVE-$\cup\uparrow$ | FiVE-$\cap\uparrow$ | Acc.$\uparrow$ |
| `LIVEditor` (full-attention) | 49.00 | 51.00 | 23.23 | 40.21 | 42.38 |
| `LIVEditor` (ISA) | **57.00** | **57.00** | **29.29** | **40.72** | **44.49** |

we concatenate the source and target videos side-by-side and feed the combined sequence into Vision-Language Models (VLMs) to assess whether the two videos remain consistent. The system prompt for the VLMs is shown in Fig. 14. In addition, to collect high-quality video editing data sets, we utilize DOVER (Wu et al., 2023a), VideoReward (Liu et al., 2025b), and VLMs to obtain samples with best visual aesthetics, and prompt-following ability.

**Post-Processing & Latent Encoding.** We first input the selected source video (raw video), target video, source video prompt and target video prompt into the VLMs to obtain the target video instruction. Simultaneously, these pairs are processed through the Resize & Resample stage, where we resize the video resolution and resample frames to keep consistency. Finally, the source video, the target video and the target video instruction are processed through a VAE (Wan 2.1 VAE (WanTeam et al., 2025)) and the text encoders to encode them into the latent, producing the final training-ready data triplets.

## C.2. Data Distribution

To evaluate the diversity and coverage of our synthesized dataset, we provide a detailed analysis of the editing task distribution, as illustrated in Fig. 12. The dataset comprises over 1.7M high-quality video-to-video editing pairs, systematically categorized into 7 primary editing modalities: **Style Transfer**, **Object Swap**, **Object Addition**, **Object Stylization**, **Object Removal**, **Human Edit** (comprising **Avatar Transformation** and **Face Detail Replacement**) and Other (comprising **Motion and Viewpoint edits**).

The distribution is characterized by a strategic balance between global transformations and localized semantic manipulations. **Style Transfer** represents the largest portion of the dataset with 33.87%. Besides these common editing tasks, a significant highlight of our training dataset is the fine-grained **Human Edit** category, which comprises 12.80%. This category is further bifurcated into **Avatar Transformation** and **Face Detail Replacement**, providing critical data for evaluating complex human priors and identity-preserving manipulations. Finally, although smaller in volume, the **Other** category (1.09%) encompasses high-complexity tasks such as **Motion Edit** and **Viewpoint Change**. essential for assessing the model's understanding of 3D geometry and temporal dynamics. This hierarchical distributio ensures that models trained on our synthesized data exhibit robust generalization across the full spectrum of video editing requirements (Li et al., 2026; Liu et al., 2024; Shao et al., 2023; 2025; Shitong Shao, 2026; Yi et al., 2025).

## C.3. Data Scheduling Process

As illustrated in Fig. 15, our training pipeline consists of two distinct stages. In the first stage, we construct a pre-training dataset by aggregating approximately 1 million synthetic samples with 0.7 million samples filtered from the Ditto dataset (Bai et al., 2025). This collection intentionally includes a mixture of high- and low-quality data. While the model demonstrated strong performance on most tasks following this stage, it exhibited limitations in object removal. Consequently, the second stage focuses on fine-tuning with a curated high-quality dataset. We generated 0.06 million samples using Minimax Remover (Zi et al., 2025a), selecting the top 0.03 million based on Dover (Wu et al., 2023a) scores. These were combined with rigorously filtered samples from public datasets (Ditto (Bai et al., 2025), LoVoRA (Xiao et al., 2025), ReCo (Zhang et al., 2025c)) and our internal data to yield a final fine-tuning set of 0.089 million samples.

*Table 9.* **FiVE-Bench Results (Color Alteration).** `LIVEditor` (ISA) outperforms `LIVEditor` (full-attn) on almost all metrics.

| Method | Structure | Background Preservation | | | | Text Align | IQA | | Temp. Consis. |
|---|---|---|---|---|---|---|---|---|---|
| | Dist.↓ | PSNR↑ | LPIPS↓ | MSE↓ | SSIM↑ | CLIP-S↑ | CLIP-S↑ | NIQE↓ | Motion Fid.↑ |
| `LIVEditor` (full-attention) | 71.53 | 15.34 | 337.03 | 474.83 | 51.52 | 27.47 | 22.32 | **6.30** | 59.38 |
| `LIVEditor` (ISA) | **65.17** | **15.91** | **300.33** | **379.59** | **54.18** | **28.22** | **22.58** | 6.42 | **60.70** |

*Table 10.* **FiVE-Bench Results (Color Alteration).** `LIVEditor` (ISA) outperforms `LIVEditor` (full-attn) on almost all metrics.

| Method | FiVE Evaluation | | | | |
|---|---|---|---|---|---|
| | YN↑ | MC↑ | FiVE-∪↑ | FiVE-∩↑ | Acc.↑ |
| `LIVEditor` (full-attention) | 83.00 | 84.00 | 73.74 | 78.35 | 79.77 |
| `LIVEditor` (ISA) | **86.00** | **90.00** | **74.76** | **81.96** | **83.18** |

# D. Additional Analysis

Here, we present visualizations and empirical analyses that could not be included in the main paper due to page limitations.

## D.1. Analysis of $||Q(K - K^c)^\top||_\infty^2$ in Theorem 3.1

Theorem 3.1 establishes that the Taylor error $\mathcal{E}_i$ is upper-bounded by terms involving $||Q(K - K^c)^\top||_\infty^2$ and $M_i$. The former term characterizes the mean intra-block variance, while the latter captures the variance between block means. Empirical analysis (Figs. 6 and D.1) indicates that the bound is dominated by $M_i$ rather than the intra-block variance term. Furthermore, as calculating the intra-block variance is computationally prohibitive, we adopt $M_i$, referred to as sharpness, as the effective proxy metric (Shao et al., 2026a;b; Shitong Shao, 2026).

# E. Additional Experimental Result

## E.1. Evaluation on FiVE-Bench

*Evaluation on Object Replacement (Rigid).* As illustrated in Table 5, we evaluate the performance of `LIVEditor-14B` (ISA) and `LIVEditor-14B` (full-attn) on the Object Replacement (Rigid) task. The results indicate that `LIVEditor-14B` (ISA) achieves superior structural consistency and background preservation. Specifically, it reduces the Structure Distance to 71.37 compared to 81.31 for the full-attention baseline. Furthermore, `LIVEditor-14B` (ISA) attains a PSNR of 15.40 and a Motion Fidelity of 58.48. These metrics surpass the baseline scores of 14.52 and 55.73 by significant margins.

**FiVE Evaluation on Object Replacement (Rigid).** Table 6 presents the results using FiVE evaluation metrics for the rigid object replacement task. `LIVEditor-14B` (ISA) consistently outperforms the full-attention counterpart across all dimensions. Notably, it achieves an Accuracy of 50.71 and a FiVE-Union score of 37.88. These values represent a substantial improvement over the `LIVEditor-14B` (full-attn) performance of 43.80 and 25.86 respectively. This demonstrates the enhanced capability of ISA in executing precise rigid object edits.

**Evaluation on Object Replacement (Non-Rigid).** We further analyze the performance on the challenging Object Replacement (Non-Rigid) task in Table 7. `LIVEditor-14B` (ISA) demonstrates remarkable stability in maintaining background details while performing complex deformations. It achieves a Structure Distance of 81.78 and an LPIPS score of 319.22. In comparison, `LIVEditor-14B` (full-attn) yields scores of 90.55 and 359.51. Additionally, the Motion Fidelity of `LIVEditor-14B` (ISA) reaches 57.99 which exceeds the 54.59 achieved by the baseline.

**FiVE Evaluation on Object Replacement (Non-Rigid).** As illustrated in Table 8, the FiVE metrics further confirm the superiority of `LIVEditor-14B` (ISA) in non-rigid scenarios. The model obtains a Yes-No (YN) score of 57.00 and a FiVE-Intersection score of 40.72. These results significantly surpass the baseline scores of 49.00 and 40.21. The overall Accuracy also sees an improvement from 42.38 to 44.49. This indicates that ISA effectively handles the complexities of non-rigid object replacement.

**Evaluation on Color Alteration.** Table 9 details the comparative results for the Color Alteration task. `LIVEditor-14B` (ISA) exhibits exceptional performance in preserving the structural integrity of the video while accurately modifying colors.

*Table 11.* **FiVE-Bench Results (Material Modification).** **LIVEditor** (ISA) outperforms **LIVEditor** (full-attn) on almost all metrics.

| Method | Structure | Background Preservation | | | | Text Align | IQA | | Temp. Consis. |
|---|---|---|---|---|---|---|---|---|---|
| | Dist.↓ | PSNR↑ | LPIPS↓ | MSE↓ | SSIM↑ | CLIP-S↑ | CLIP-S↑ | NIQE↓ | Motion Fid.↑ |
| **LIVEditor** (full-attention) | 81.75 | 14.18 | 386.07 | 551.62 | 45.65 | 26.07 | 21.38 | **5.94** | 56.55 |
| **LIVEditor** (ISA) | **73.09** | **15.34** | **310.98** | **429.40** | **52.35** | **27.04** | **21.94** | 6.27 | **59.27** |

*Table 12.* **FiVE-Bench Results (Object Addition).** **LIVEditor** (ISA) outperforms **LIVEditor** (full-attn) on almost all metrics.

| Method | Structure | Background Preservation | | | | Text Align | IQA | | Temp. Consis. |
|---|---|---|---|---|---|---|---|---|---|
| | Dist.↓ | PSNR↑ | LPIPS↓ | MSE↓ | SSIM↑ | CLIP-S↑ | CLIP-S↑ | NIQE↓ | Motion Fid.↑ |
| **LIVEditor** (full-attention) | 62.98 | 14.57 | 344.83 | 394.66 | 41.89 | 24.63 | 21.16 | **0.67** | 57.05 |
| **LIVEditor** (ISA) | **54.79** | **16.19** | **276.89** | **299.82** | **47.59** | **25.49** | **21.48** | 0.78 | **76.39** |

It achieves a PSNR of 15.91 and a SSIM of 54.18. Conversely, LIVEditor-14B (full-attn) records lower values of 15.34 and 51.52. Moreover, LIVEditor-14B (ISA) secures a Text Alignment score of 28.22 which outperforms the baseline score of 27.47.

**FiVE Evaluation on Color Alteration.** We present the FiVE evaluation for Color Alteration in Table 10. LIVEditor-14B (ISA) demonstrates robust performance with a Model Consistency (MC) score of 90.00 and a FiVE-Union score of 74.76. These figures represent a clear advantage over the LIVEditor-14B (full-attn) results of 84.00 and 73.74. The overall Accuracy of LIVEditor-14B (ISA) stands at 83.18 compared to 79.77 for the baseline. This confirms the effectiveness of ISA in precise color editing tasks.

**Evaluation on Material Modification.** Table 11 displays the comparative analysis for the Material Modification task. The results indicate that LIVEditor-14B (ISA) excels in preserving fine-grained textural details while adhering to editing prompts. It achieves a Structure Distance of 73.09 and an LPIPS score of 310.98. These represent significant improvements over the full-attention baseline scores of 81.75 and 386.07. Furthermore, LIVEditor-14B (ISA) demonstrates superior background preservation with a PSNR of 15.34 compared to 14.18. The model also exhibits enhanced Text Alignment and Motion Fidelity reaching scores of 27.04 and 59.27 respectively.

**Evaluation on Object Addition.** We report the performance on the Object Addition task in Table 12. LIVEditor-14B (ISA) demonstrates a remarkable ability to seamlessly integrate new objects into existing scenes. It achieves a significantly lower MSE of 299.82 compared to the 394.66 recorded by LIVEditor-14B (full-attn). The Structure Distance also improves from 62.98 to 54.79. Notably, LIVEditor-14B (ISA) shows a substantial advantage in temporal dynamics achieving a Motion Fidelity score of 76.39. This far exceeds the baseline performance of 57.05. Additionally, the model attains a higher PSNR of 16.19 and a CLIP-S score of 25.49.

### E.2. Evaluation on IVE-Bench

Table 13 presents a comprehensive quantitative comparison on IVE-Bench. We benchmark LIVEditor-14B (ISA) and LIVEditor-14B (full-attention) against eight state-of-the-art methods including Ditto, InsV2V, LucyEdit, and VACE. The results indicate that LIVEditor-14B (ISA) achieves a leading Total score of 0.67. This performance matches the best-performing baselines while exhibiting a more robust and balanced profile across all evaluation dimensions.

In terms of Video Quality Metrics, LIVEditor-14B (ISA) demonstrates exceptional superiority. It secures the highest scores across all four sub-metrics with 0.97 for Subjective Quality, 0.98 for Background Preservation, 0.99 for Flickering, and a perfect 1.00 for Motion. These results outperform strong competitors like Ditto and InsV2V. Notably, the sparse attention mechanism effectively eliminates temporal jitter and artifacts.

Regarding Instruction Compliance, LIVEditor-14B (ISA) significantly improves upon the LIVEditor-14B (full-attention) baseline. The Instruction score rises from 0.43 to 0.47. Furthermore, it achieves a VTSS of 0.040 and a Quantity score of 0.40. This performance surpasses widespread methods such as InsV2V and VACE which score only 0.39 and 0.25 in Instruction respectively. It indicates that our model accurately interprets and executes complex editing prompts.

Finally, within the Video Fidelity Metrics, our model maintains high consistency despite the sparsification of attention. It achieves a Fidelity score of 0.76 and a Semantic consistency of 0.91. These scores are comparable to the full-attention baseline and exceed those of methods like AnyV2V and StableV2V. Overall, LIVEditor-14B (ISA) offers the most

*Table 13.* **IVE-Bench Evaluation Results.** Our proposed `LIVEditor` (ISA) achieved superior performance across 7 metrics, establishing it as the leading algorithm overall.

| Method | Dimension Performance | | | | Video Quality Metrics | | | | Instruction Compliance Metrics | | | | | Video Fidelity Metrics | | |
|---|---|---|---|---|---|---|---|---|---|---|---|---|---|---|---|---|
| | Total↑ | Qual.↑ | Instr.↑ | Fid.↑ | Subj.↑ | Back.↑ | Flick.↑ | Motion↑ | VTSS↑ | O.Sem.↑ | P.Sem.↑ | Satis.↑ | Qty.↑ | Sem.↑ | Motion↑ | Cont.↑ |
| Ditto | 0.67 | 0.78 | **0.49** | 0.73 | 0.96 | 0.98 | 0.97 | 0.99 | 0.038 | **0.25** | **0.24** | **3.87** | 0.30 | 0.89 | 0.79 | 3.64 |
| InsV2V | 0.67 | 0.80 | 0.39 | 0.82 | 0.94 | 0.96 | 0.97 | 0.97 | 0.045 | 0.24 | 0.23 | 3.06 | 0.30 | 0.95 | 0.86 | **4.05** |
| LucyEdit | 0.64 | **0.82** | 0.34 | 0.75 | 0.95 | 0.96 | 0.98 | 0.99 | 0.051 | 0.24 | 0.22 | 2.84 | 0.20 | 0.93 | 0.68 | 3.83 |
| VACE | 0.63 | 0.80 | 0.25 | **0.83** | 0.95 | 0.97 | 0.98 | 0.98 | 0.045 | 0.24 | 0.22 | 2.16 | 0.20 | **0.97** | **0.89** | 4.03 |
| ICVE | 0.60 | 0.71 | 0.45 | 0.64 | 0.95 | 0.97 | **0.99** | **1.00** | 0.017 | 0.23 | 0.23 | 3.62 | 0.30 | 0.85 | 0.46 | 3.55 |
| Omni-Video | 0.59 | 0.78 | 0.44 | 0.54 | 0.96 | 0.97 | 0.98 | 0.99 | 0.038 | 0.22 | 0.23 | 3.36 | **0.40** | 0.81 | 0.51 | 2.85 |
| AnyV2V | 0.58 | 0.73 | 0.42 | 0.59 | 0.89 | 0.94 | 0.97 | 0.97 | 0.026 | 0.22 | 0.24 | 3.34 | 0.30 | 0.80 | 0.82 | 2.75 |
| StableV2V | 0.51 | 0.69 | 0.43 | 0.41 | 0.85 | 0.92 | 0.96 | 0.96 | 0.019 | 0.20 | 0.24 | 3.56 | 0.20 | 0.70 | 0.75 | 1.79 |
| **LIVEditor (full-attention)** | 0.66 | 0.77 | 0.43 | 0.76 | 0.96 | 0.97 | 0.98 | 0.99 | 0.035 | 0.23 | 0.23 | 3.58 | 0.20 | 0.91 | 0.83 | 3.77 |
| **LIVEditor (ISA)** | **0.67** | 0.79 | 0.47 | 0.76 | **0.97** | **0.98** | **0.99** | **1.00** | **0.040** | 0.24 | 0.23 | 3.64 | **0.40** | 0.91 | 0.81 | 3.78 |

favorable trade-off between visual quality, instruction adherence, and content fidelity among all evaluated methods.

### E.3. Evaluation on VIE-Bench

Table 14 presents a detailed quantitative evaluation on VIE-Bench covering five distinct editing tasks. In the Object Addition task, `LIVEditor-14B` (ISA) demonstrates exceptional capability with an average score of 8.84. This performance significantly outperforms the previous state-of-the-art method Omni-Video which achieves 6.24. Similarly, in the Object Swapping scenario, our model attains an average score of 8.14. It surpasses the competitive generative baseline Pika which scores 7.41. Regarding the Object Removal task, `LIVEditor-14B` (ISA) obtains scores of 8.57 for Background Preservation and 6.33 for Visual Quality. These specific metrics exceed those of baselines such as InsV2V and VACE. For Video Stylization, `LIVEditor-14B` (ISA) establishes a new benchmark with an average score of 8.16. This represents a substantial improvement over InsV2V and Omni-Video. Finally, in the challenging Hybrid task which involves composite instructions, `LIVEditor-14B` (ISA) maintains robust performance with an average score of 8.12 compared to 5.43 for Omni-Video. Overall, `LIVEditor-14B` (ISA) consistently matches or surpasses the `LIVEditor-14B` (full-attention) baseline across these diverse categories (Bai et al., 2026).

## F. The Proof of Error Bound of 0-th Order Taylor Approximation

We aim to rigorous derive the relationship between the approximation error and the attention sharpness using a first-order perturbation analysis.

*Proof.* **Step 1: Second-Order Analysis via Taylor's Theorem with Integral Remainder.** To achieve a rigorous bound without vague residuals, we expand the softmax $\sigma(z)$ around the approximate logits $\tilde{z}_i$. By Taylor's theorem:

$$\Delta p_i = J(\tilde{z}_i)\Delta z_i + \int_0^1 (1-t)H(\tilde{z}_i + t\Delta z_i)[\Delta z_i \otimes \Delta z_i]dt \tag{6}$$

where $H$ is the Hessian tensor. Since the entries of the softmax Hessian are bounded by a constant (e.g., $\|H\|_\infty \leq 1$), the second term is bounded by $\frac{1}{2}\mathcal{C}_H\|\Delta z_i\|_2^2$. Taking the $L_2$ norm of $\Delta p_i$:

$$\|\Delta p_i\|_2^2 \leq \|J(\tilde{z}_i)\|_2^2 \cdot \|\Delta z_i\|_2^2 + \mathcal{C}_H\|\Delta z_i\|_2^4 \tag{7}$$

This establishes a strict value-level error bound for each token.

**Step 2: Linking Jacobian Stability to Sharpness.** The spectral norm of the Jacobian is $\|J\|_2 = p_{\max}(1 - p_{\max})$. We observe that $M_u \approx \sum p_k^2$. For the active operating range of diffusion models (where distributions are concentrated but not yet collapsed into a one-hot delta function), the term $p_{\max}(1 - p_{\max})$ is positively correlated with the distribution energy $\|p\|_2^2$. Specifically, using the property $p_{\max}^2 \leq \|p\|_2^2$, and noting that for $p_{\max} \in (0, 0.5]$, $\|J\|_2$ is strictly increasing, and even for $p_{\max} > 0.5$, $\|J\|_2^2$ remains bounded by $\|p\|_2^2$. We thus adopt the conservative upper bound $\|J(\tilde{z}_i)\|_2^2 \leq \|S_2(i)\|_2^2$ to maintain the direct link to our sharpness metric $M_u$.

**Step 3: Precise Lipschitz Decomposition of the Interaction Term.** Let $f(i) = \|S_2(i)\|_2^2$ and $g(i) = \|\Delta z_i\|_2^2$. The Lipschitz constant $L$ of $f(i)$ is well-defined: $\nabla f = 2\sum p_k(J\nabla z)$, which is bounded by $2\|W_Q\|\|W_K\|$. By the definition

*Table 14.* **VIE-Bench Evaluation Results.** Our proposed `LIVEditor` (ISA) achieves state-of-the-art performance across all tasks, including Add, Remove, Swap, Style, and Hybrid.

| Task | Method | Model Type | | VIE-Bench Score | | | |
|---|---|---|---|---|---|---|---|
| | | Comm. | Base | Follow↑ | Pres.↑ | Qual.↑ | Avg.↑ |
| **Add** | Omni-Video | ✓ | ✓ | 5.70 | 6.14 | 6.29 | 6.24 |
| | InsV2V | ✓ | ✓ | 3.55 | 5.89 | 3.40 | 4.28 |
| | VACE | ✓ | ✗ | 3.94 | 6.70 | 3.93 | 4.85 |
| | **LIVEditor (full-attn)** | ✓ | ✓ | 7.77 | 8.83 | 8.43 | 8.40 |
| | **LIVEditor (ISA)** | ✓ | ✓ | **8.87** | **9.07** | **8.60** | **8.84** |
| **Swap** | Pika | ✗ | ✓ | 7.54 | 7.85 | 6.84 | 7.41 |
| | Omni-Video | ✓ | ✓ | 4.73 | 4.86 | 4.66 | 4.75 |
| | InsV2V | ✓ | ✓ | 5.30 | 6.43 | 4.97 | 5.57 |
| | VACE | ✓ | ✗ | 6.17 | 7.55 | 6.20 | 6.64 |
| | **LIVEditor (full-attn)** | ✓ | ✓ | **7.91** | 8.23 | 7.91 | 8.02 |
| | **LIVEditor (ISA)** | ✓ | ✓ | 7.77 | **8.71** | **7.94** | **8.14** |
| **Remove** | Omni-Video | ✓ | ✓ | 6.00 | 5.97 | 4.81 | 5.59 |
| | InsV2V | ✓ | ✓ | 1.21 | 3.77 | 1.32 | 2.10 |
| | VACE | ✓ | ✗ | 1.81 | 3.88 | 2.36 | 2.68 |
| | MiniMax | ✓ | ✗ | **6.96** | 7.52 | 6.04 | **6.84** |
| | DiffuEraser | ✓ | ✗ | 6.35 | 6.81 | 5.58 | 6.24 |
| | **LIVEditor (full-attn)** | ✓ | ✓ | 5.43 | 8.12 | 6.28 | 6.61 |
| | **LIVEditor (ISA)** | ✓ | ✓ | 5.55 | **8.57** | **6.33** | 6.81 |
| **Style** | Omni-Video | ✓ | ✓ | 5.49 | 4.66 | 5.96 | 5.37 |
| | InsV2V | ✓ | ✓ | 7.84 | 8.09 | 6.44 | 7.45 |
| | **LIVEditor (full-attn)** | ✓ | ✓ | 7.96 | 8.16 | 7.48 | 7.87 |
| | **LIVEditor (ISA)** | ✓ | ✓ | **8.06** | **8.46** | **7.96** | **8.16** |
| **Hybrid** | Omni-Video | ✓ | ✓ | 5.44 | 5.07 | 5.77 | 5.43 |
| | InsV2V | ✓ | ✓ | 5.03 | 5.97 | 4.97 | 5.32 |
| | **LIVEditor (full-attn)** | ✓ | ✓ | 8.07 | 7.76 | 7.68 | 7.84 |
| | **LIVEditor (ISA)** | ✓ | ✓ | **8.10** | **8.19** | **8.08** | **8.12** |

of $L$-Lipschitz continuity, for any $i$ in a block of diameter $\delta$, the deviation from the mean $M_u$ is $|f(i) - M_u| \le L\delta$. We decompose the block-level expectation:

$$\mathbb{E}[f(i)g(i)] = \mathbb{E}[(M_u + (f(i) - M_u))g(i)] \tag{8}$$

$$\le M_u\mathbb{E}[g(i)] + \mathbb{E}[|f(i) - M_u| \cdot g(i)] \tag{9}$$

$$\le (M_u + L\delta)\mathbb{E}[g(i)] \tag{10}$$

Combining Step 1 and Step 3, we obtain the final tightened bound:

$$\mathcal{E}_u \le (M_u + L\delta) \cdot \mathbb{E}[\|\Delta z_i\|_2^2] + \mathcal{O}(\mathbb{E}[\|\Delta z_i\|_2^4]) \tag{11}$$

The proof is complete. □ □

## G. The Algorithm Implementation of ISA

In this section, we present the pseudocode implementation of ISA. For the sake of clarity, we assume identical sequence lengths and head dimensions for $\mathbf{Q}$, $\mathbf{K}$, and $\mathbf{V}$. We provide the pseudocode for ISA in Algorithm 1 and for Block-Wise 0-th Taylor Sparse Attention in Algorithm 2. In the latter, we omit batch size and head dimensions to simplify the presentation.

## H. Implementation of 0-th Taylor Sparse Attention: Triton Vs. TileLang

We implemented the forward and backward passes of Block-Wise zeroth-order Taylor Sparse Attention using Triton to train `LIVEditor-14B`. Additionally, we developed a TileLang-based (Wang et al., 2025) version for comparison. Fig 17 illustrates the relative speedup of the TileLang implementation versus the Triton version on a Hopper GPU. We observe that

```
Role & Expertise
You are a specialized AI editor with expertise in visual-textual consistency management for video content.
Your primary function is to analyze edited still images and corresponding edit instructions, \
then generate updated long prompts that accurately reflect the modifications while preserving \
the original video's narrative essence and stylistic qualities.
Core Objectives
    Visual-Textual Alignment: Ensure the updated prompt precisely describes the edited image's content
    Context Preservation: Maintain the original prompt's narrative flow, emotional tone, and compositional structure
    Edit-Specific Adaptation: Tailor modifications to the specific type of edit performed \
 (removal, addition, swap, stylization, style transfer, or motion effect)
    Semantic Consistency: Ensure all modifications maintain logical coherence within the scene context
Operational Framework
Input Analysis Phase
    Original Prompt Assessment:
        Identify key elements: subjects, actions, environment, style descriptors, lighting conditions
        Analyze narrative structure: scene setting, action sequencing, emotional tone
        Note technical specifications: resolution, aspect ratio, artistic style mentions
    Edit Instruction Interpretation:
        Categorize edit type: removal/addition/swap/stylization/style/motion
        Identify target objects and affected elements
        Understand spatial relationships and scope of changes
    Visual Analysis:
        Examine edited image for all modifications
        Verify edit instruction compliance
        Note any secondary effects or unintended changes
Modification Strategy
    Element Mapping: Create mental correspondence between visual elements and text descriptions
    Change Isolation: Identify exact phrases requiring modification
    Context Integration: Ensure new elements blend naturally with existing description
    Quality Validation: Verify grammatical correctness and narrative flow
Edit-Type Specific Guidelines
Object Removal:
    Eliminate references to removed objects
    Adjust spatial descriptions if necessary
    Maintain scene balance in description
Object Addition:
    Insert new elements in contextually appropriate positions
    Add relevant descriptors: size, color, position, state
    Ensure logical placement within existing elements
Object Swap:
    Replace specific object references
    Maintain consistent descriptive complexity
    Update any object-specific interactions
Object Stylization:
    Modify appearance descriptors only
    Preserve all other object characteristics
    Maintain environmental consistency
Style Transfer:
    Update style descriptors while preserving content
    Adjust lighting/color descriptions to match new style
    Maintain original composition and elements
Motion Edit:
    Modify action verbs and motion descriptors
    Add appropriate motion qualifiers (speed, direction, intensity)
    Update temporal elements if necessary
Viewpoint Change:
    Update camera perspective descriptors (aerial view, eye-level, low-angle, etc.)
    Adjust spatial relationships between objects based on new viewpoint
    Modify compositional elements to reflect changed perspective
    Maintain scene content while altering viewing angle

Output Specifications
    Format: Single line containing complete updated prompt
    Style: Maintain original writing style and technical specificity
    Length: Match original prompt's descriptive density
    Structure: Preserve original narrative flow and element sequencing
Quality Assurance Checklist
✓ All visual elements accurately described
✓ Edit instructions fully implemented
✓ Original style and tone preserved
✓ Grammatical and syntactic correctness
✓ Logical scene coherence maintained
✓ Technical specifications consistent
✓ Narrative flow uninterrupted
Examples
Object Removal:
    Original: "A bustling medieval market square with merchants selling goods,\
 children playing near the fountain, and horses tied to wooden posts"
    Instruction: "Remove all horses from the scene"
    Output: Updated Long Prompt: "A bustling medieval market square with merchants selling goods and \
 children playing near the fountain"
Object Addition:
    Original: "A peaceful library interior with wooden shelves, reading tables, \
and stained glass windows"
    Instruction: "Add a gray cat sleeping on one of the reading tables"
    Output: Updated Long Prompt: "A peaceful library interior with wooden shelves, \
reading tables featuring a gray cat sleeping on one, and stained glass windows"
Object Swap:
    Original: "A modern kitchen with stainless steel appliances, marble countertops, \
and contemporary pendant lighting"
    Instruction: "Swap the contemporary pendant lighting with vintage brass chandeliers"
    Output: Updated Long Prompt: "A modern kitchen with stainless steel appliances, \
marble countertops, and vintage brass chandeliers"
Object Stylization:
    Original: "A golden retriever playing in autumn leaves in a forest clearing"
    Instruction: "Make the dog appear as a detailed charcoal sketch"
    Output: Updated Long Prompt: "A golden retriever rendered as a detailed charcoal sketch,\
 playing in autumn leaves in a forest clearing"
Style Transfer:
    Original: "A digital photograph of a New York City street at night with yellow taxis and neon signs"
    Instruction: "Convert to 1950s film noir cinematic style"
    Output: Updated Long Prompt: "A 1950s film noir cinematic scene of a New York City street at night \
 with yellow taxis and neon signs, featuring high contrast black \
and white photography and dramatic shadow play"
Motion Edit:
    Original: "A ballet dancer standing en pointe in a sunlit studio with mirrored walls"
    Instruction: "Make the dancer spinning rapidly with motion blur on the tutu"
    Output: Updated Long Prompt: "A ballet dancer spinning rapidly en pointe in a sunlit studio \
 with mirrored walls, with motion blur emphasizing the movement of the tutu"
Viewpoint Change:
    Original: "A street-level view of a bustling city intersection with \
pedestrians crossing and traffic lights changing"
    Instruction: "Change to a bird's-eye view looking down on the scene"
    Output: Updated Long Prompt: "A bird's-eye view of a bustling city intersection with \
 pedestrians crossing and traffic lights changing,\
 seen from above with buildings surrounding the crossing"
Execution Protocol
    Receive input: original long prompt, edited image, edit instruction
    Process through analysis and modification phases
    Generate updated prompt adhering to all guidelines
    Output only the final result in specified format
    Do not include any commentary or additional information
Now ready to process input and generate updated long prompts.
```

*Figure 13.* **One of the system prompt we use in Target Frame Generation phase.**

TileLang is more efficient when the No Sparsity Ratio is low and the sequence length is short. However, as the No Sparsity Ratio and sequence length increase, the Triton implementation outperforms TileLang. Given that video editing typically involves sequence lengths of approximately 64K (corresponding to a $544 \times 960$ resolution on Wan-2.1/2.2), we employ the Triton version for our comparative analysis with other sparse attention methods.

## I. Additional Explanation

**Note on MLLM-based and Concurrent Works.** We exclude comparisons with recent MLLM-based methods, such as EditVerse (Ju et al., 2025), UniVideo (Wei et al., 2025), and InstructX (Mou et al., 2025), due to the unavailability of public implementations and their reliance on non-standardized, proprietary benchmarks. Furthermore, these approaches operate under a distinct paradigm: while they leverage the massive representational capacity of MLLMs, our work specifically targets the acceleration of video editing efficiency. Finally, we do not include LoVoRA (Xiao et al., 2025) and ReCo (Zhang et al., 2025c) as they are concurrent pre-prints that have not yet appeared in peer-reviewed venues

**Analysis of Performance Gain over Full Attention.** ISA operates as an approximation algorithm that reconstructs full attention via context token selection and block-wise zeroth-order Taylor sparse attention. Contrary to the expectation that approximation implies degradation, ISA outperforms standard full attention after training. We attribute this improvement to the effective pruning of irrelevant noise tokens. In video editing tasks based on in-context learning, context tokens serve primarily as conditional priors rather than direct generation targets. This role parallels feature extraction in discriminative tasks and relies on high semantic density. The TopK selection mechanism explicitly filters for semantic relevance and discards uninformative tokens to maintain a high-quality context window. Furthermore, the block-wise Taylor attention

```
"""Role: You are an expert Video Quality Assessment (VQA) agent specializing in Spatio-Temporal Consistency.
Input: You will be provided with multiple stitched frames, each containing two sequences side-by-side (Left vs. Right).
Task: Evaluate the structural and motion consistency between the left and right videos.
Evaluation Criteria (Focus on these):
1) Global Scene Geometry: Do the background structures align spatially?
2) Camera Motion & Trajectory: Do both sides share the same pan/tilt/zoom/static behavior?
3) Global Optical Flow: Does the overall motion direction and speed match?
Exclusion Criteria (Ignore these):
- Do NOT penalize object identity or style changes if motion stays consistent.
- Do NOT penalize local texture artifacts unless they break physical logic.
Scoring Rubric:
1 = Completely Inconsistent; unrelated scenes or conflicting camera motion.
2 = Low Consistency; major geometric misalignments or motion conflicts.
3 = Partially Consistent; background roughly matches with noticeable sliding artifacts.
4 = High Consistency; geometry and motion stable, only minor boundary jitter. Only assign 4 when motion and geometry stay tightly aligned with minimal residual drift.
5 = Perfect Consistency; motions feel from the same 3D world and are indistinguishable in trajectory quality.
Calibration Rule: Be conservative. Reserve scores 4 or 5 for only the strongest ~20% of samples; downgrade borderline cases to 3 or below. If unsure between 3 vs 4, pick 3.
Output Constraint: Return ONLY a single integer (1-5). Do not provide explanations."""
```

*Figure 14.* **The system prompt we use in Target Video Generation phase.**

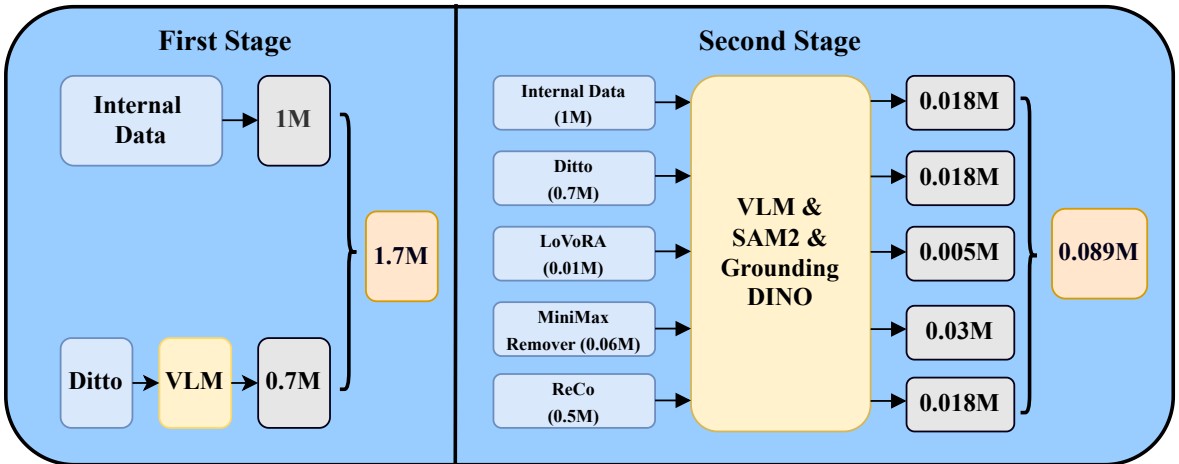

*Figure 15.* The data scheduling strategy for **LIVEditor** proceeds in two stages. In the first stage, we train on a mixture of high-quality and lower-quality data. In the second stage, we curate a dataset of 0.06M samples generated by Minimax Remover, which is then combined with our proprietary data and three open-source datasets. After extensive filtering, we obtain a final set of 0.089M high-quality samples for fine-tuning

introduces a distinct attention paradigm rather than functioning merely as a mathematical approximation. As illustrated in Fig. 9, the model actively adapts to this sparsity during training. Within this framework, the local mean approximation implicitly serves as a secondary mechanism for information filtering. We conclude that the superior performance of ISA stems from the synergy between TopK pre-selection and Taylor-based aggregation, as both components effectively isolate critical semantic information.

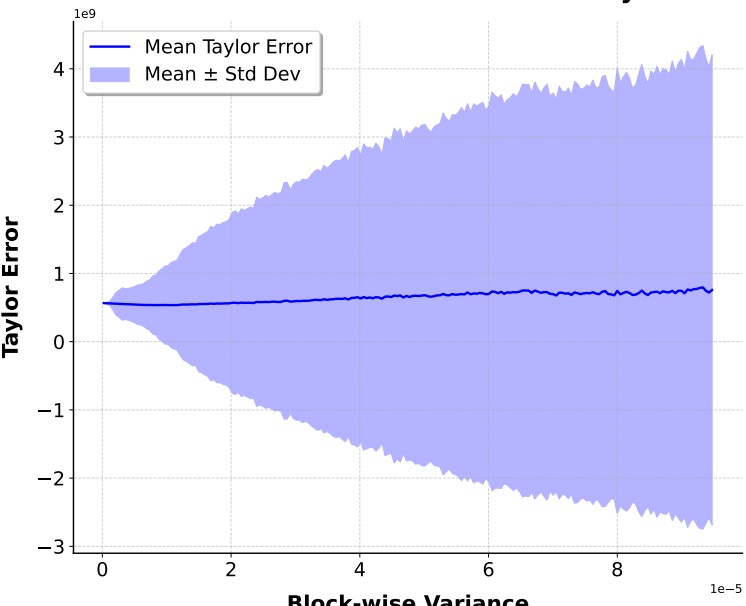

*Figure 16.* Visualization results demonstrate that the Taylor approximation error $\mathcal{E}_i$ exhibits negligible correlation with the block variance term $||Q(K - K^c)^\top||_\infty^2$. (Note that the magnitudes are inflated due to the absence of regularization; Fig. 6 in the main text presents the results following regularization). Consequently, $||Q(K - K^c)^\top||_\infty^2$ fails to serve as a reliable indicator for the magnitude of $\mathcal{E}_i$.

---

**Algorithm 1** Pseudocode for In-Context Sparse Attention (ISA).

---

1: **Input:** Query $Q$, Key $K$, Value $V \in \mathbb{R}^{B \times H \times S \times D}$.
2: **Input:** Source length $L_{src}$, Context length $L_{ctx}$, Block size $b$, No Sparsity Ratio $\alpha_{ns}$, Flat Ratio $\alpha_f$, Select Ratio $\alpha_s$, (Optional) Coarse-grained attention stacking factor $\gamma$.
3: **Output:** Attention Output $O \in \mathbb{R}^{B \times H \times S \times D}$.
4: **Stage 1: Block-wise Compression & Coarse Scoring**
5: Pad $Q, K, V$ to length divisible by $b$. Total blocks $T = S_{pad}/b$.
6: $Q^c = \text{mean}(Q_{\text{blocks}}); K^c = \text{mean}(K_{\text{blocks}}); V^c = \text{mean}(V_{\text{blocks}});$          // Compress to [B, H, T, D]
7: $S_{\text{coarse}} = Q^c(K^c)^\top;$          // Low-res attention map [B, H, T, T]
8: $O_{\text{coarse}} = S_{\text{coarse}} V^c;$          // Coarse output for residual connection
9: **Stage 2: Context Selection (KV Construction)**
10: Score context blocks: $s_{\text{ctx}} = \sum_i S_{\text{coarse}}[:, :, : L_{src}/b, L_{src}/b :][:, :, i, :];$          // Sum scores along Q dim
11: $\mathcal{I}_{\text{topk}} = \text{TopK}(s_{\text{ctx}}, k = \alpha_s \cdot \text{int}(L_{ctx}/b));$          // Select most relevant context blocks
12: $K_{\text{selected}} = \text{Gather}(K, \mathcal{I}_{\text{topk}}); V_{\text{selected}} = \text{Gather}(V, \mathcal{I}_{\text{topk}});$
13: $K_{\text{new}} = \text{Concat}(K[: L_{src}], K_{\text{selected}});$          // Concat source KV and selected context KV
14: $V_{\text{new}} = \text{Concat}(V[: L_{src}], V_{\text{selected}});$
15: **Stage 3: Sharpness-Aware Query Splitting**
16: *// Calculate sharpness for each Query block based on attention distribution*
17: $S_{\text{var}} = \text{Var}(S_{\text{coarse}}[:, :, :, : L_{src}/b], \text{dim=-1});$
18: Sort blocks by $S_{\text{var}}$: $\mathcal{P} = \text{Argsort}(S_{\text{var}}, \text{descending});$
19: Split indices: $\mathcal{I}_{\text{sharp}} = \mathcal{P}[: T - \text{int}(\alpha_f T)], \mathcal{I}_{\text{flat}} = \mathcal{P}[T - \text{int}(\alpha_f T) :];$          // $1$-$\alpha_f$ Split
20: $Q_{\text{sharp}} = \text{Gather}(Q, \mathcal{I}_{\text{sharp}});$
21: $Q_{\text{flat}} = \text{Gather}(Q, \mathcal{I}_{\text{flat}});$
22: **Stage 4: Decoupled Kernel Execution**
23: $O_{\text{flat}} = \text{Block-Wise 0-th Taylor Sparse Attention}(Q_{\text{flat}}, K_{\text{new}}, V_{\text{new}}, \alpha_{ns}(\frac{L_{src} + \alpha_s \text{int}(L_{ctx}/b) b}{b}));$ // Use Block-wise 0-th Taylor Kernel
24: $O_{\text{sharp}} = \text{FlashAttention}(Q_{\text{sharp}}, K_{\text{new}}, V_{\text{new}});$          // Use standard FA for flat blocks
25: **Stage 5: Reconstruction**
26: Initialize $O_{\text{out}} = \mathbf{0};$
27: $\text{Scatter}(O_{\text{out}}, \mathcal{I}_{\text{sharp}}, O_{\text{sharp}});$          // Put sharp results back
28: $\text{Scatter}(O_{\text{out}}, \mathcal{I}_{\text{flat}}, O_{\text{flat}});$          // Put flat results back
29: $O_{\text{final}} = O_{\text{out}}[: S] + \gamma \cdot O_{\text{coarse}};$          // Add residual from Stage 1
30: **return** $O_{\text{final}};$

---

---

**Algorithm 2** Implementation of Block-wise 0-th Taylor Sparse Attention.

---

1: **Input:** Matrices $Q, K, V \in \mathbb{R}^{N \times d}$ (BF16), block size $B$, sparsity count $k$.
2: **Output:** Attention Output $O \in \mathbb{R}^{N \times d}$.
3: Divide $Q, K, V$ into blocks $Q_i, K_j, V_j$ of size $B \times d$. Total blocks $T_q = N/B, T_{kv} = N/B$.
4: **Stage 1: Pre-computation (Block Reductions)**
5: **for** $i = 1$ **to** $T_q$ **do**
6:    $Q_i^c = \mathrm{mean}(Q_i, \mathrm{axis} = 0)$;                                    // Reduce Q to [T_q, d]
7: **end for**
8: **for** $j = 1$ **to** $T_{kv}$ **do**
9:    $K_j^c = \mathrm{mean}(K_j, \mathrm{axis} = 0)$;   $V_j^c = \mathrm{mean}(V_j, \mathrm{axis} = 0)$;              // Reduce KV to [T_kv, d]
10: **end for**
11: **Stage 2: Topology Selection**
12: $S_{\mathrm{coarse}} = Q^c(K^c)^\top$;                                          // Compute block-level relevance score
13: $\mathcal{I} = \mathrm{TopK}(S_{\mathrm{coarse}}, k, \dim = -1)$;                        // Select indices $\mathcal{I}_i$ for each Query block $i$
14: **Stage 3: Fused Attention Kernel**
15: **for** $i = 1$ **to** $T_q$ **do**
16:    Load $Q_i$ into SRAM; Load $\mathcal{I}_i$ (indices for $Q_i$);
17:    Initialize online softmax stats: $m_i = -\infty, \ell_i = 0, O_i = 0$;
18:    *// Part A: Exact Sparse Attention (FlashAttention)*
19:    **for** $j \in \mathcal{I}_i$ **do**
20:       Load $K_j, V_j$ into SRAM;
21:       $S_{ij} = Q_i K_j^\top \cdot \mathrm{scale}$;
22:       $m_{\mathrm{new}} = \max(m_i, \mathrm{rowmax}(S_{ij}))$;
23:       $P_{ij} = \exp(S_{ij} - m_{\mathrm{new}})$;
24:       $\ell_i = e^{m_i - m_{\mathrm{new}}} \ell_i + \mathrm{rowsum}(P_{ij})$;
25:       $O_i = e^{m_i - m_{\mathrm{new}}} O_i + P_{ij} V_j$;
26:       $m_i = m_{\mathrm{new}}$;
27:    **end for**
28:    *// Part B: 0-th Taylor Interval Approximation*
29:    **for** $j = 1$ **to** $T_{kv}$ **do**
30:       **if** $j \notin \mathcal{I}_i$ **then**
31:          Load $K_j^c, V_j^c$ into SRAM;                                // Load block means
32:          $S_{ij}^{\mathrm{avg}} = Q_i (K_j^c)^\top \cdot \mathrm{scale}$;                           // Approximate score
33:          $m_{\mathrm{new}} = \max(m_i, \mathrm{rowmax}(S_{ij}^{\mathrm{avg}}))$;
34:          $P_{ij}^{\mathrm{avg}} = \exp(S_{ij}^{\mathrm{avg}} - m_{\mathrm{new}})$;
35:          *// Update accumulators weighted by block size B*
36:          $\ell_i = e^{m_i - m_{\mathrm{new}}} \ell_i + \mathrm{rowsum}(P_{ij}^{\mathrm{avg}}) \cdot B$;
37:          $O_i = e^{m_i - m_{\mathrm{new}}} O_i + (P_{ij}^{\mathrm{avg}} V_j^c) \cdot B$;
38:          $m_i = m_{\mathrm{new}}$;
39:       **end if**
40:    **end for**
41:    $O_i = \mathrm{diag}(\ell_i)^{-1} O_i$;                                           // Final Normalization
42:    Store $O_i$ to HBM;
43: **end for**
44: **return** $O$;

---

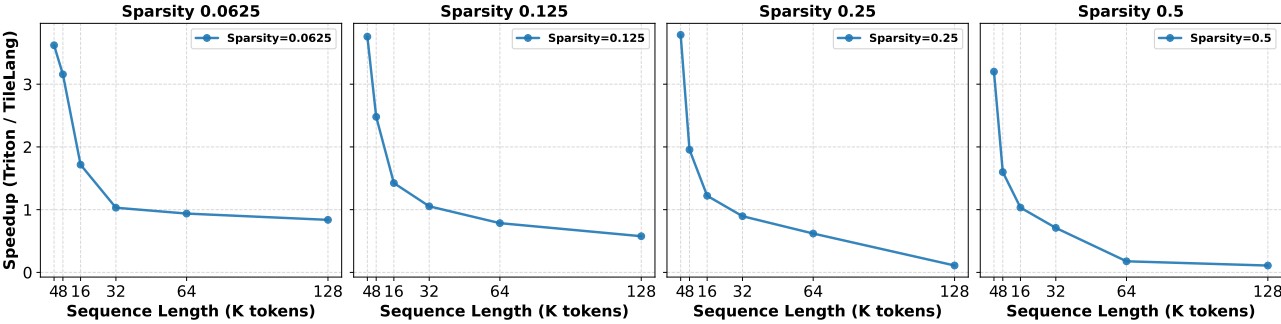

*Figure 17.* **Efficiency Comparison on Hopper GPU**: TileLang vs. Triton Implementations of Block-Wise Zeroth-Order Taylor Sparse Attention.

