# OpenReview forum: "LIVEditor-14B: Lightning Unified Video Editing via In-Context Sparse Attention"
_ICML.cc/2026/Conference — ICML 2026 regular_

### Official Review · Reviewer_mj4G · 2026-03-10

**Soundness:** 3
**Presentation:** 3
**Significance:** 3
**Originality:** 2
**Overall Recommendation:** 5
**Confidence:** 4

**Summary:**

This paper approaches in-context learning–based video editing from the perspective of sparse attention, and proposes In-Context Sparse Attention to reduce the computational cost of long-sequence video editing while preserving editing quality.

**Compliance With Llm Reviewing Policy:**

Affirmed.

**Final Justification:**

The rebuttal has mainly addressed my concerns. Therefore, I recommend this paper for acceptance.

**Key Questions For Authors:**

I am inclined to increase my score if the authors can address the following concerns (listed in order of importance):

1.Could the authors provide some comparison results with non-in-context video editing models?

2.The authors are also encouraged to include a visualization of the video results in the rebuttal by presenting selected frames (or video clips), so that the model’s video editing capability can be better demonstrated.

3.In addition, the authors may consider providing a more refined bound analysis (optional).

**Limitations:**

Yes, the author discusses the potential negative social impacts that video editing models may cause, as they are highly sensitive to privacy.

**Strengths And Weaknesses:**

**Strengths**

1.The authors first analyze statistical characteristics of the attention distributions and provide a well-motivated rationale for their Pre-Selection mechanism.

2.They further propose Block-Wise 0-th Order Taylor Sparse Attention, which constitutes a structurally simple and direct sparsification strategy.

3.The paper provides both Triton and TileLang implementations and conducts extensive experiments on EditVerseBench and other benchmarks. Multiple sparse attention baselines are carefully compared, offering comprehensive empirical validation of the proposed method.

4.The data pipeline is well described, and the data distribution is clearly analyzed, which is a strength of the paper.

**Weaknesses**

1.However, the description of the editing model itself is relatively brief. The detailed presentation of the sparse mechanism occupies substantial space, which comes at the expense of a more thorough explanation of the editing framework. The authors are encouraged to provide a more detailed description of the editing model architecture, particularly clarifying how different conditioning is used.

2.In addition, the necessity of the in-context learning formulation deserves further discussion. While the paper focuses on improving efficiency under the in-context setting, it does not examine whether in-context conditioning is inherently superior to a cross-conditioning alternative. It would be informative to understand the performance trade-offs and efficiency implications of using a cross-conditioning design instead.


3.Regarding Theorem 3.1, the proof appears somewhat simplified. The linearization and local smoothness assumptions play a central role in the derivation, but are somehow too strong. The result serves more as a theoretical justification rather than a rigorous proof. Moreover, the provided bound is relatively loose and does not offer a tight characterization of the approximation error.

4.Finally, as a video editing method, the paper primarily presents single-frame visualizations. The absence of multi-frame visualizations or accompanying video results somewhat limits the reviewer’s ability to fully assess temporal consistency and editing stability.

---

> ### Author Rebuttal · Authors · 2026-03-30
>
> _Q1. Clarify LIVEditor architecture and conditioning._
>
> > ISA is the core contribution, while LIVEditor is the framework used to validate it. We kept its description concise because the backbone is standard. The main points are:
> >
> > 1. **Backbone & ICL.** Built on Wan2.2, LIVEditor concatenates source/context video latents with instruction tokens into one sequence and processes them jointly with shared attention blocks.
> > 2. **Conditioning & RoPE.** We use decoupled RoPE, encoding source and context groups independently to reduce positional bias and better handle variable sequence lengths.
> > 3. **Scope.** ISA only replaces the attention computation; the overall conditioning interface and training recipe remain standard and are detailed in Appendix B.
>
> _Q2. Explain whether ICL is genuinely better than cross-conditioning._
>
> > To clarify that ICL is not merely unnecessary overhead, we provide the following evidence:
> >
> > 1. During the rebuttal, we trained a cross-conditioning (VACE) baseline on our dataset for 5 days (24*5 hours) across 4 nodes. As shown below, ICL significantly outperforms cross-conditioning in all metrics.
> > 2. The advantages of ICL for video tasks have been substantiated by InstructX [1], UniVideo [2], and EditVerse [3].
>
> | Method (Wan2.2) | Quality | Text Align | Temporal Consistency | Editing Quality | Pick Score (Frame) | Pick Score (Video) |
> | --- | --- | --- | --- | --- | --- | --- |
> | VACE | 5.61 | 18.63 |. 21.39 | 15.82 | 96.58 | 96.39 |
> | ISA (Ours) | 7.89 | 20.09| 27.19| 24.55 | 99.32| 99.22|
>
> > [1]. InstructX: Towards Unified Visual Editing with MLLM Guidance
> > [2]. UniVideo: Unified Understanding, Generation, and Editing for Videos
> > [3]. EditVerse: Unifying Image and Video Editing and Generation with In-Context Learning
>
> _Q3. Strengthen the proof of Theorem 3.1._
>
> > We strengthen Theorem 3.1 by using a second-order Taylor expansion with integral remainder, clarifying the relation between $\|J\|_2$ and $M_u$, and formalizing local smoothness with a Lipschitz assumption.
>
> **Theorem.** Let $S_1(i) = \sigma(z_i)$ and $S_2(i) = \\sigma(\\tilde{z}\_i)$ be the true and approximate attention distributions for token $i \\in Block\_u$. Let $M_u \triangleq \mathbb{E}_{i \in \text{Block}_u} [\|S_2(i)\|_2^2]$ be the **block-wise sharpness metric**. Assuming the attention energy $f(i) = \|S_2(i)\|_2^2$ is $L$-Lipschitz within the block, where $L$ is determined by the spectral norms of projection weights $\|W_Q\|, \|W_K\|$, the expected approximation error $\\mathcal{E}\_u = \\mathbb{E}\_{i \\in \\text{Block}\_u}[\\|S\_1(i) - S\_2(i)\\|_2^2]$ is bounded by:
>
> $$
> \mathcal{E}\_u \le (M_u + L\delta) \cdot \mathbb{E}_{i \in \text{Block}_u} [\|\Delta z_i\|_2^2] + \mathcal{C}_H \mathbb{E}[\|\Delta z_i\|_2^4]
> $$
>
> where $\delta$ is the block diameter and $\mathcal{C}_H$ is a constant related to the maximum curvature (Hessian) of the softmax function.
>
> **Proof.**
>
> **Step 1.** Expand $\sigma(z)$ around $\tilde{z}_i$:
>
> $$
> \Delta p_i = J(\tilde{z}_i) \Delta z_i + \int_0^1 (1-t) H(\tilde{z}_i + t\Delta z_i) [\Delta z_i \otimes \Delta z_i] dt
> $$
>
> With bounded Hessian entries (e.g., $\|H\|_\infty \le 1$), we obtain:
>
> $$
> \|\Delta p_i\|_2^2 \le \|J(\tilde{z}_i)\|_2^2 \cdot \|\Delta z_i\|_2^2 + \mathcal{C}_H \|\Delta z_i\|_2^4
> $$
>
> **Step 2.** The spectral norm of the Jacobian is $\|J\|\_2 = p\_{\max}(1 - p_{\max})$. We use $M_u \approx \sum p_k^2$. For the operating range of diffusion models, $p_{\max}(1 - p_{\max})$ is positively correlated with $\|p\|\_2^2$. Using $p_{\max}^2 \le \|p\|\_2^2$, and noting that for $p_{\max} \in (0, 0.5]$, $\|J\|\_2$ is increasing, while for $p_{\max} > 0.5$, $\\|J\\|\_2^2$ remains bounded by $\|p\|_2^2$, we adopt the conservative upper bound $\|J(\tilde{z}_i)\|_2^2 \le \|S_2(i)\|_2^2$ to link the bound to $M_u$.
>
> **Step 3.** Let $f(i) = \|S_2(i)\|_2^2$ and $g(i) = \|\Delta z_i\|_2^2$. The Lipschitz constant $L$ of $f(i)$ is well-defined since $\nabla f = 2 \sum p_k (J \nabla z)$ is bounded by $2 \|W_Q\| \|W_K\|$. For any $i$ in a block of diameter $\delta$, $|f(i) - M_u| \le L\delta$, so
>
> $$
> \begin{aligned}
> \mathbb{E}[f(i)g(i)] &= \mathbb{E}[(M_u + (f(i) - M_u)) g(i)] \\
> &\le M_u \mathbb{E}[g(i)] + \mathbb{E}[|f(i) - M_u| \cdot g(i)] \\
> &\le (M_u + L\delta) \mathbb{E}[g(i)]
> \end{aligned}
> $$
>
> Combining Steps 1 and 3 gives:
>
> $$
> \mathcal{E}_u \le (M_u + L\delta) \cdot \mathbb{E}[\|\Delta z_i\|_2^2] + \mathcal{O}(\mathbb{E}[\|\Delta z_i\|_2^4])
> $$
>
> The proof is complete. $\square$
>
> _Q4. Provide multi-frame visualizations or video results._
>
> > To address the limitations of single-frame comparisons, we have provided a video-based demo in the supplementary link (https://liveditors.github.io/liveditor_demo1/). Regarding Fig. 18 (Row 1, Col 3), we apologize for the drafting error: the label should be "Add object" (specifically, a cherry tomato), which correctly reflects the intended edit.

---

> > ### Author Rebuttal · Reviewer_mj4G · 2026-04-03
> >
> > The authors have adequately addressed my questions.

---

> > > ### Author Response · Authors · 2026-04-03
> > >
> > > Thank you for your response. We are glad to have resolved the raised issues and further strengthened our work.

---

### Official Review · Reviewer_QiAz · 2026-03-11

**Soundness:** 2
**Presentation:** 3
**Significance:** 3
**Originality:** 2
**Overall Recommendation:** 3
**Confidence:** 4

**Summary:**

This paper proposes **In-Context Sparse Attention (ISA)** for unified video editing under an ICL-style formulation where source and context tokens are concatenated and processed jointly. The method has two main components: (i) a **pre-selection stage** that prunes low-saliency context tokens using pooled coarse attention and Top-K block selection, and (ii) a **grouped computation stage** that routes “sharp” queries to full attention while sending “flat” queries to a block-wise **0-th order Taylor sparse attention** approximation. On top of this attention mechanism, the authors build **LIVEditor**, a video editing system trained with a large-scale data pipeline and a two-stage training recipe. Empirically, the paper reports strong results on multiple video-editing benchmarks, along with meaningful latency improvements over full attention and better quality than several sparse-attention baselines.

**Compliance With Llm Reviewing Policy:**

Affirmed.

**Key Questions For Authors:**

**How exactly does the token flattening order relate to the underlying (time, height, width) structure during pre-selection?**

Since ISA forms blocks along the flattened sequence dimension and then averages tokens within each block, neighboring tokens inside a block need not be neighbors in actual spatiotemporal layout. Please clarify whether this is an intended design choice, whether ISA should be viewed as operating purely in flattened token space, and what implications this has for the semantic quality of the coarse block representation and Top-K selection.

*Why this matters:*

a convincing answer would improve my confidence in the soundness of the pre-selection mechanism.

**Why is sequence concatenation treated as the default conditioning interface, and did you consider alternatives such as channel concatenation or other fusion schemes?**

The current paper assumes an ICL-style source/context concatenation setup and then sparsifies attention within that setup. Please discuss whether ISA is solving an unavoidable problem, or primarily a problem induced by one particular modeling choice.

*Why this matters:*

if the authors can justify why sequence concatenation is preferable or unavoidable in their setting, it would strengthen both significance and soundness.

**Limitations:**

Yes, but only partially. The authors appropriately acknowledge misuse risks such as misleading content and note dataset safety filtering, which is valuable; however, the discussion would be stronger if it also explicitly addressed key technical limitations of ISA - such as its operation in flattened token space, its dependence on sequence concatenation, and potential failure cases under aggressive sparsification - along with more concrete residual risks of deceptive or unauthorized video editing.

**Strengths And Weaknesses:**

**Strengths**

The paper addresses a real and important bottleneck in modern video editing systems: the quadratic cost of attention under long video sequences, especially when source and context are concatenated in an ICL-style setup. The overall method is coherent: the paper first motivates that context tokens are less salient than source tokens, then uses this observation to prune context, and finally adds an adaptive query routing mechanism based on a sharpness proxy linked to Taylor approximation error. This gives the method a reasonably clear conceptual arc rather than presenting ISA as a purely heuristic engineering stack. The empirical section is also a strong point: the authors report results on several benchmarks, compare against both full-attention and sparse-attention baselines, provide latency measurements, and include ablations over the main sparsity hyperparameters.

**Weaknesses**

The pre-selection design operates on non-overlapping blocks formed along the flattened sequence dimension, followed by block-wise mean pooling and Top-K context selection. However, the paper does not explicitly discuss how this flattening relates to the underlying spatial and temporal structure of video tokens. As a result, tokens grouped into the same block may be adjacent in sequence while not being adjacent in space or time in the original video, which makes the semantic meaning of the coarse block representation unclear. This is an important limitation that should be stated explicitly, since ISA’s pre-selection is performed in flattened token space rather than with an explicitly spatiotemporal-aware grouping.

The paper treats ICL-style sequence concatenation of source and context tokens as a fixed design assumption rather than a modeling choice to be justified. The motivation and method are built entirely around the claim that recent video editing systems concatenate source and context tokens and then apply full attention, after which ISA prunes and re-organizes the resulting long sequence. However, the paper does not discuss alternative conditioning strategies — most notably channel concatenation or other fusion schemes — that may avoid the sequence-length blow-up altogether instead of sparsifying attention after the fact. As a result, the contribution is narrower than it may first appear: it proposes an efficient sparse-attention solution within one specific ICL formulation, but does not evaluate whether that formulation itself is preferable to other conditioning interfaces for video editing.

---

> ### Author Rebuttal · Authors · 2026-03-30
>
> _Q1. Explain why flattened-space block pooling does not destroy spatiotemporal semantics._
>
> > We apologize for the notation in Algorithm 1, which simplified the implementation for clarity. In practice, our input $Q, K, V$ are pre-partitioned into structured $4 \times 4 \times 4$ spatiotemporal blocks (similar to https://github.com/hao-ai-lab/FastVideo/blob/main/fastvideo/attention/backends/video_sparse_attn.py L24-40), following the standard procedures in STA and VSA. In other words, prior to being fed into ISA, the input $Q$, $K$, and $V$ tensors are pre-partitioned into tiles based on spatiotemporal proximity—much like in STA and VSA. Since this approach is a standard practice employed by nearly all sparse attention mechanisms in video generation, we did not explicitly detail this aspect in our paper; we sincerely apologize for this omission. Consequently, the "flattened space" refers only to a hardware-friendly memory layout rather than a disruption of spatiotemporal adjacency.
>
> _Q2. Justify the ICL sequence-concatenation design against channel concatenation alternatives._
>
> > We appreciate the reviewer’s insight. To clarify that ICL is a superior architectural choice rather than a source of "unnecessary overhead," we provide the following evidence:
> >
> > 1. During the rebuttal, we trained a channel concatenation baseline on our dataset for 5 days (5*24hours) across 4 nodes. As shown in the table below, ICL significantly outperforms the channel concatenation baseline in all metrics.
> > 2. The advantages of ICL for video tasks have been substantiated by prominent works such as InstructX [1], UniVideo [2], and EditVerse [3].
> > 3. Regarding the experiments on VACE (cross-attention/controlnet), please refer to our response to reviewer mj4G.
>
> | Method (Wan2.2) | Quality | Text Align | Temporal Consistency | Editing Quality | Pick Score (Frame) | Pick Score (Video) |
> | --- | --- | --- | --- | --- | --- | --- |
> | Channel Concatenation  | 5.69 | 18.88 |  22.43 | 18.15 | 96.57 | 97.12 |
> | ISA (Ours) | 7.89 | 20.09| 27.19| 24.55 | 99.32| 99.22|
>
> > [1]. InstructX: Towards Unified Visual Editing with MLLM Guidance
> > [2]. UniVideo: Unified Understanding, Generation, and Editing for Videos
> > [3]. EditVerse: Unifying Image and Video Editing and Generation with In-Context Learning

---

> > ### Author Rebuttal · Reviewer_QiAz · 2026-04-04
> >
> > I thank the authors for the thorough rebuttal.
> > - Regarding Q1, the clarification that tokens are pre-partitioned into spatiotemporal blocks prior to ISA (following standard practice in STA/VSA) fully addresses my concern. The issue was indeed one of notation and exposition rather than a fundamental design flaw. I would recommend making this explicit in the revised paper (e.g., a brief note in Section X or Algorithm 1).
> > - Regarding Q2, the newly trained channel concatenation baseline provides direct empirical evidence that ICL-style concatenation is not merely a convenience but a meaningfully better design choice for this setting. The consistent improvement across all metrics is convincing. Combined with the references to prior work validating ICL for video tasks, this satisfactorily justifies the architectural choice.
> >
> > Given that both of my main concerns are resolved, I am willing to raise my score accordingly.

---

> > > ### Author Response · Authors · 2026-04-04
> > >
> > > We sincerely thank the reviewer for the constructive feedback. In the revised manuscript, we will explicitly clarify that tokens are pre-partitioned into spatiotemporal blocks prior to the application of ISA. We also appreciate the recognition of our algorithm's merits and the effort invested in providing these suggestions.
> > >
> > > We are encouraged that our responses have addressed all your concerns and that you have expressed a willingness to raise the score. We would highly appreciate it if you could manually adjust the "Overall Recommendation" (by using the "Edit" button) score accordingly to reflect our discussion.

---

### Official Review · Reviewer_rGUN · 2026-03-11

**Soundness:** 3
**Presentation:** 3
**Significance:** 3
**Originality:** 3
**Overall Recommendation:** 4
**Confidence:** 3

**Summary:**

This paper proposes In-context Sparse Attention (ISA), the first experimentally lossless sparse framework tailored for ICL video editing. They have two important empirical observations. First, the context tokens exhibit significantly lower saliency than source tokens. Second, they theoretically prove and empirically validate that Query sharpness correlates with approximation error. Motivated by these findings, ISA implements an efficient pre-selection strategy to prune redundant context, followed by a dynamic query grouping mechanism to route between full attention and sparse attention. Extensive experiments demonstrate that LIVEditor achieves a 60% reduction in latency across various benchmarks without compromising visual fidelity.

**Compliance With Llm Reviewing Policy:**

Affirmed.

**Final Justification:**

The authors have addressed most of my concerns. I have raised the score.

**Key Questions For Authors:**

Please see the weaknesses and provide more evidence and experiments.

**Limitations:**

Please see the weaknesses.

**Strengths And Weaknesses:**

Strongness:

1. The paper is well motivated by the two empirical observations. The method is reasonable and  useful.
2. The paper is well written and easy to follow.
3. The experiment results demonstrate that LIVEditor achieves an obvious speedup without compromising visual fidelity.

Weaknesses:

1. Are the empirical observations robust across various editing models? The observations are only derived from the customized LIVEditor which is continually pretrained on Wan 2.2. I worry about that the observations may be different in other models.
2.  Are the empirical observations related to the specific design of two seperate RoPE? Could you please give me more evidence and experiments?
3. There are many hyperparameters that require tuning. Besides, there are no ablation experiments on $L_Q$ and $L_K$. In my opinion, the pooling size may have a great impact on the attention distribution and the final performance.
4. It would be better to provide the visualization comparison between original full attention and ISA to see the influence of ISA.
5. Could the ISA be compatible with SVG[1] or other acceleration techiniques, like AdaCache, DuCa, Taylorseer, TeaCache.

[1] Sparse VideoGen: Accelerating Video Diffusion Transformers with Spatial-Temporal Sparsity

---

> ### Author Rebuttal · Authors · 2026-03-30
>
> _Q1. Cross-model generalization._
>
> > To verify robustness, we conducted additional experiments on HunyuanVideo (hyvideo-1.5) using 4 H100 nodes for 24*5 hours. Although training has not yet fully converged within the rebuttal window, ISA already consistently outperforms the Full-Attention baseline across most metrics (e.g., 7.04 vs. 6.89) while delivering a 1.39x training speedup. We will include converged curves in the final version.
>
> | Method (HYVideo-1.5) | Quality | Text Align | Temporal Consistency | Editing Quality | Pick Score (Frame) | Pick Score (Video) | SpeedUp |
> | --- | --- | --- | --- | --- | --- | --- | --- |
> | Full-Attention | 6.89 | 19.42 | 26.93 | 22.15 | 98.02 | 97.92 | 1.00 |
> | ISA (Ours) | 7.04 | 19.46 | 26.89 | 22.43 | 98.16 | 98.06 | 1.39 |
>
> _Q2. Dual-RoPE effect._
>
> > 1. Context Token Importance and RoPE Design. The observed low importance of context tokens is not a byproduct of the dual-RoPE module design. In a training-free setting, switching from two separate RoPE modules to a single module caused context-token importance to decrease further, suggesting the observation is not a RoPE artifact.
>
> | Block Index | Source Token x Source Token | Source Token x Context Token (two seperate RoPE) | Source Token x Context Token (one RoPE) |
> | --- | --- | --- | --- |
> | 0 | 0.0012823498 | 0.0009398725 | 0.0008998068 |
> | 1 | 0.0018618515 | 0.0003603707 | 0.0003076054 |
> | 2 | 0.0018669190 | 0.0003553033 | 0.0003182571 |
> | 3 | 0.0016730642 | 0.0005491579 | 0.0005296185 |
> | 4 | 0.0015613183 | 0.0006609039 | 0.0006062208 |
> |...|...|...|...|
> | 39 | 0.0018297837 | 0.0003924385 | 0.0003463757 |
>
> > 2. Justification of the RoPE Variant. This RoPE variant was recently adopted in VideoCoF (CVPR 2026) [1]. By encoding source and target videos independently, it can extrapolate to previously unseen lengths.
> >
> > [1]. VideoCoF: Unified Video Editing with Temporal Reasoner, CVPR2026.
>
> _Q3. Ablation on $b$ and $L\_{src}$._
>
> > We provide ablation results for block size $b$ and source sequence length $L_{src}$. Increasing $b$ improves inference speed but gradually lowers quality; we select $b=64$ as the best trade-off. The impact of $L_{src}$ is minor, likely because most EditVerseBench samples are 3s-5s and close to our training scale.
>
> | Block Size (Wan2.2) | 256 | 64 (Ours) | 32 | 8 |
> | --- | --- | --- | --- | --- |
> | $b$ | [8,8,4] | [4,4,4] | [4,4,2] | [2,2,2] |
> | Quality | 7,69 | 7.85 | 7.89 | 7.91 |
> | SpeedUp | 1.72x | 1.47x | 1.23x | 0.29x |
>
> | $L_{src}$ (Wan2.2) | 26520 | 34680 (Ours) | 42840 |
> | --- | --- | --- | --- |
> | Quality | 7.87 | 7.89 | 7.88 |
> | SpeedUp | 1.47x | 1.47x | 1.47x |
>
> _Q4. Visual comparison._
>
> > Following your suggestion, we have provided a side-by-side visualization comparison between Full-Attention and ISA in our anonymous demo: [https://liveditors.github.io/liveditor_demo2/]. The results show that ISA consistently achieves slightly better visual quality than the Full-Attention baseline, particularly in motion consistency and detail preservation.
>
> _Q5. TeaCache compatibility._
>
> > We provide results for combining ISA with TeaCache [1], a training-free temporal caching method designed for DiTs. ISA is fully compatible with TeaCache and achieves significant additive speedups.
>
> | Setting (Wan2.2) | Only ISA | ISA + TeaCache ($\delta = 0.1$) | ISA + TeaCache ($\delta = 0.15$) | ISA + TeaCache ($\delta = 0.2$) |
> | --- | --- | --- | --- | --- |
> | Text Align | 20.09 | 20.10 | 20.05 | 19.98 |
> | Quality | 7.89 | 7.78 | 7.67 | 7.44 |
> | SpeedUp | 1.47x | 2.69x | 3.62x | 4.87x |
>
> > At $\delta=0.1$, ISA+TeaCache achieves a 2.69x speedup with nearly lossless performance (7.89 vs. 7.78 quality). We will include this study in the final version.
> >
> > [1] TeaCache: Timestep Embedding Tells: It's Time to Cache for Video Diffusion Model

---

> > ### Author Rebuttal · Reviewer_rGUN · 2026-04-01
> >
> > The authors provide additional experiments and address my concerns.

---

> > > ### Author Response · Authors · 2026-04-03
> > >
> > > We appreciate your continued feedback. We are pleased to have addressed the issues you raised, which has helped us further refine our work.

---

### Official Review · Reviewer_C4cr · 2026-03-13

**Soundness:** 2
**Presentation:** 2
**Significance:** 3
**Originality:** 3
**Overall Recommendation:** 4
**Confidence:** 3

**Summary:**

This paper studies the computational bottleneck of in-context learning (ICL) video editing It proposes In-context Sparse Attention (ISA), built on two insights: most context tokens have low saliency, and a block-level query sharpness correlates with 0-th order Taylor attention approximation error. ISA therefore first prunes context KV blocks, then routes high-error ''sharp'' queries to full attention and low-error ''flat'' ones to a computationally efficient Taylor sparse attention. On top of this, the paper builds LIVEditor, a lightning and unified video editing model trained in two stages on a large synthetic+public dataset, and reports strong results and about 1.47× end-to-end speedup over full attention on EditVerseBench.

**Compliance With Llm Reviewing Policy:**

Affirmed.

**Final Justification:**

I understand that temporal fidelity is a broader challenge and not unique to this work. Still, I think it would strengthen the paper to discuss the possible reasons behind this phenomenon, especially since temporal fidelity is part of the paper’s stated evaluation scope. I see the contribution of ISA, although I remain somewhat hesitant about some of the demos.

**Key Questions For Authors:**

Please see the weaknesses above. I'm happy to raise my rating if the rebuttal addresses my concerns.

**Limitations:**

No. The paper does not discuss its limitations.

**Strengths And Weaknesses:**

**Strengths:**
- S1. The paper targets a real and important problem in ICL video editing.
- S2. The ISA is technically sound. While heuristic, it is reasonably well supported by analysis and experiments, i.e., the source-vs-context attention distribution, and the sharpness correlates with Taylor approximation error.
- S3. Experimental results show that ISA is strong among the compared sparse attention baselines while retaining a meaningful speedup.

**Weaknesses:**
- W1. ISA assumes that most context tokens are low-saliency and can be pruned and supports this mainly with average attention statistics. This is suggestive, but it does not fully establish that subtle, yet important edit cues are preserved in complex scenarios. In video editing, object insertion/removl, fine localized edits, etc. may depend exactly on the kind of subtle context cues that receive little average attention, and are therefore easy to underestimate with average statistics. I would love to see more analysis on this point.
- W2. For a video editing paper, the current evidence remains insufficiently. While reporting benchmark metrics, I am not fully persuaded that this is adequate for assessing real-world video editing quality, particularly regarding temporal coherence and instruction following. These aspects are difficult to judge from the current single-frame output, especially given the lack of corresponding text instructions. In addition, in some examples, the actual edit effect is hard to verify, e.g., the swap object in Fig 18 (row 1, col 3), the background change in Fig 18 (last row, first col) and the add object in Fig 19 (last row, first col).
- W3. ISA is governed by three sparsity hyperparameters, and the ablation shows it is highly sensitive to the Flat Ratio. The default values ($ \alpha_f=0.5, \alpha_{ns}=0.0625, \alpha_s=0.125$) seem carefully hand-chosen, which raises some concern that the reported quality-speed tradeoff depends on a favorable tuning regime rather than robust performance across different settings.

---

> ### Author Rebuttal · Authors · 2026-03-30
>
> _Q1. Provide deeper analysis showing that pruning low-saliency context tokens does not hurt fine-grained edits._
>
> > We appreciate the insightful concern regarding fine-grained edits. We have added new comparison cases (e.g., object insertion/removal, style transfer) at https://liveditors.github.io/liveditor_demo2/ to address this.
> >
> > 1. Empirical Evidence: Visualizations show that ISA matches or even exceeds Full-Attention in detail preservation and localized editing. We observed no performance degradation when handling subtle context cues.
> > 2. Theoretical Justification: While ISA prunes many tokens per layer, our 40-layer WAN backbone ensures robustness. Since the specific tokens retained vary dynamically across layers, their collective "union" effectively covers all spatio-temporal regions throughout the network. This multi-layer diversity prevents permanent information loss, ensuring precise editing capabilities across all video areas.
> >
> > We will include this analysis and additional qualitative examples in the revised version.
>
> _Q2. Provide multi-frame/video evidence, clarify hard-to-verify edit cases, and include the corresponding text instructions for the visual results._
>
> > To address the limitations of single-frame comparisons, we have provided a video-based demo in the supplementary link (https://liveditors.github.io/liveditor_demo1/). Regarding Fig. 18 (Row 1, Col 3), we apologize for the drafting error: the label should be "Add object" (specifically, a cherry tomato), which correctly reflects the intended edit.
> >
> > We appreciate the reviewer’s suggestion regarding the clarity of instruction following. We will include full text prompts for all visual results in the revised version. In the meantime, comprehensive prompts and complete video sequences are available at https://liveditors.github.io/liveditor_demo1/ and https://liveditors.github.io/liveditor_demo2/.
>
> _Q3. Clarify whether ISA's hyperparameters are overly hand-tuned, and show robustness across settings/tasks._
>
> > We agree that the **Flat Ratio** is the most sensitive ISA hyperparameter, because it most directly controls the quality-speed trade-off. To address the concern that our default setting may be overly hand-tuned, we provide two complementary pieces of evidence:
> >
> > 1. **Within-model robustness on Wan2.2.** We swept the Flat Ratio on EditVerseBench while keeping the other ISA settings fixed. The results show a **smooth trade-off curve rather than a brittle single-point optimum**. In particular, there is a broad high-quality region around `0.25-0.75`: the overall Quality varies only from `7.64` to `7.77`, Text Align from `19.93` to `20.00`, Temporal Consistency from `26.84` to `27.11`, and Editing Quality from `23.67` to `23.80`. We therefore choose the default Flat Ratio `0.5` as the best operating point. Lower Flat Ratios provide higher speed, while larger Flat Ratios move closer to Full-Attention with diminishing efficiency gains, consistent with the trend already shown in Fig. 8 of the paper.
>
> | Flat Ratio (Wan2.2) | 0.0625 | 0.125 | 0.25 | 0.5 (Ours) | 0.75 |
> | --- | --- | --- | --- | --- | --- |
> | Quality | 7.29 | 7.61 | 7.64 | 7.77 | 7.66 |
> | Text Align | 19.75 | 19.94 | 19.93 | 20.00 | 19.96 |
> | Temporal Consistency | 26.37 | 26.82 | 26.84 | 27.11 | 26.98 |
> | Editing Quality | 23.18 | 23.66 | 23.67 | 23.80 | 23.79 |
> | Pick Score (Frame) | 98.89 | 99.05 | 99.20 | 99.19 | 99.18 |
> | Pick Score (Video) | 98.66 | 98.83 | 99.10 | 99.10 | 99.03 |
>
> > 2. **Cross-model robustness beyond Wan2.2.** To test whether the same sparsification strategy remains effective on another backbone, we conducted additional experiments on HunyuanVideo (hyvideo-1.5) using 4 H100 nodes for 24*5 hours. Importantly, we **directly reuse the Wan2.2-tuned ISA hyperparameters on HYVideo without model-specific retuning**. Although training has not yet fully converged within the rebuttal window, ISA already outperforms the Full-Attention baseline across most metrics (e.g., 7.04 vs. 6.89) while delivering a 1.39x training speedup. This suggests that, although the ISA hyperparameters are tuned on Wan2.2, the method is not tied to a single narrow favorable regime and transfers well to another architecture.
>
> | Method (HYVideo-1.5) | Quality | Text Align | Temporal Consistency | Editing Quality | Pick Score (Frame) | Pick Score (Video) | SpeedUp (Compare to FA) |
> | --- | --- | --- | --- | --- | --- | --- | --- |
> | Full-Attention | 6.89 | 19.42 | 26.93 | 22.15 | 98.02 | 97.92 | 1.00 |
> | ISA (Ours) | 7.04 | 19.46 | 26.89 | 22.43 | 98.16 | 98.06 | 1.39 |

---

> > ### Author Rebuttal · Reviewer_C4cr · 2026-04-02
> >
> > Thank you for providing the video demos. While reviewing the demo page (https://liveditors.github.io/liveditor_demo1/), I noticed two issues related to temporal correspondence that I would appreciate clarification on.
> >
> > First, several demo pairs show mismatched durations between the source and edited videos. For example, in the horse scene (demo 4), the source is about 4 seconds while the edited video is 5 seconds; in the highway scene (demo 8), the source is 3 seconds while the edited video is 4 seconds.  Other demos appear temporally matched in duration. For an in-context editing method that jointly processes source and context tokens, I would have expected temporal alignment between input and output to be enforced, or at least handled consistently. Could the authors clarify the source of these mismatches?
> >
> > Second, in several demos, the edited videos exhibit noticeably slower motion than the source videos. For example, in the pancake scene (demo 2), the chocolate sauce drips much more slowly in the edited output, and in the landscape scene (demo 7), the walking figure moves at a visibly reduced pace despite the matched video duration. Is this due to a frame-rate mismatch between source and target, or from the decoupled RoPE making temporal alignment inherently difficult?
> >
> > Because some demos are matched while others are not, it is unclear whether this reflects a systematic limitation of the method or simply an inconsistency in the demo presentation. In either case, to my understanding, for video editing tasks, temporal fidelity between input and output should be fundamental, and I would appreciate clarification on how temporal correspondence is handled during inference.
> >
> >
> > ----
> >
> > Thank you for the clarification. I understand that temporal fidelity is a broader challenge and not unique to this work. Still, I think it would strengthen the paper to discuss the possible reasons behind this phenomenon, especially since temporal fidelity is part of the paper’s stated evaluation scope. I'll consider raising my score. I see the contribution of ISA, although I remain somewhat hesitant about some of the demos.

---

> > > ### Author Response · Authors · 2026-04-03
> > >
> > > We sincerely thank the reviewer for the meticulous observations and the opportunity to clarify these details. We address your concerns below:
> > >
> > > 1. **Mismatched Durations**: We sincerely apologize for this oversight. The mismatched durations were not caused by algorithmic inconsistencies, but rather by errors in our demo preparation pipeline.
> > >
> > > Specifically, this occurred due to two factors:
> > >
> > > - Intermediate Evaluation Files: Our evaluation pipeline relies on YAML files that extract a specific number of frames from source videos with varying original frame rates. We mistakenly uploaded some of these intermediate files instead of the final outputs.
> > >
> > > - Accidental Frame Interpolation: During the manual cropping of the source videos (using CapCut), an automatic AI frame interpolation feature was inadvertently triggered, causing some videos to be exported at 30 FPS or 60 FPS.
> > >
> > > Fundamentally, our method maintains a strict 1-to-1 frame correspondence between the source and target videos. To correct this presentation error, we have completely re-processed and standardized all demo videos to a unified 16 FPS. The source and edited videos now have perfectly matched durations. We have updated the project page accordingly:  https://liveditors.github.io/liveditor_demo1/.
> > >
> > > 2. **Slowed Motion and Temporal Fidelity**: We openly acknowledge this limitation. The edited videos do occasionally exhibit altered motion speeds (such as the chocolate sauce or the walking figure).
> > >
> > > However, we wish to clarify that **this is not inherently a fatal flaw of the decoupled RoPE**, but rather a prevalent challenge in current video diffusion models. In our empirical testing, even state-of-the-art closed-source models with vastly larger capacities and training datasets (e.g., **Kling 3.0 Omni**) exhibit similar temporal inconsistencies during the identical editing tasks (i.e., the chocolate sauce).
> > >
> > > We respectfully emphasize that the core contribution of our work is to **explore and optimize the sparse attention mechanism within In-Context Learning (ICL) for video editing**. Our primary objective is to demonstrate that by **utilizing our ISA, trained on our dataset (primarily derived from Ditto), LIVEditor achieves superior editing quality and significantly faster inference speeds compared to full attention baselines**. While achieving perfect physical and temporal fidelity remains an open problem for the broader community, we believe our current results successfully validate the efficiency and effectiveness of ISA for visual ICL.
> > >
> > > We hope these clarifications address your concerns regarding the demo presentation and the scope of our contributions.

---

### Decision · Program_Chairs · 2026-04-30

**Decision:**

Accept (regular)

**Comment:**

This paper introduces In-context Sparse Attention (ISA) to address the computational bottleneck of in-context learning video editing. By pruning low-saliency context tokens and routing queries based on sharpness, the method significantly reduces latency without compromising visual fidelity.

The reviewers' consensus leaned toward acceptance (1 Accept, 2 Weak Accepts, 1 Weak Reject who indicated willingness to raise their score). Reviewers praised the clear motivation, the solid empirical findings, and the 60% reduction in latency. Reviewers initially raised concerns about the lack of multi-frame video results, the necessity of the in-context learning formulation compared to cross-conditioning, and the robustness of the hyperparameters.

During the rebuttal phase, the authors addressed these concerns effectively. They provided comprehensive video-based demonstrations, trained extensive baselines over 5 days to justify their architecture, and conducted additional experiments on HunyuanVideo to prove hyperparameter robustness. The reviewers found these new experiments convincing and acknowledged that their concerns were addressed. However, one reviewer remained somewhat hesitant about the visual quality of certain video demos.

The paper is technically solid, well-motivated, and provides meaningful speedups for the video editing community. Therefore, we recommend acceptance. Please ensure that the final camera-ready version incorporates all the changes promised during the rebuttal phase.

Additionally, we assure the authors that the AC has reviewed their confidential comments, and this was taken into consideration.